# THE EVOLUTION OF OUT-OF-DISTRIBUTION ROBUSTNESS THROUGHOUT FINE-TUNING

## ABSTRACT

Although machine learning models typically experience a drop in performance on out-of-distribution data, accuracies on in- versus out-of-distribution data are widely observed to follow a single linear trend when evaluated across a testbed of models. Models that are more accurate on the out-of-distribution data relative to this baseline exhibit "effective robustness" and are exceedingly rare. Identifying such models, and understanding their properties, is key to improving out-of-distribution performance. We conduct a thorough empirical investigation of effective robustness during fine-tuning and surprisingly find that models pre-trained on larger datasets exhibit effective robustness during training that vanishes at convergence. We study how properties of the data influence effective robustness, and we show that it increases with the larger size, more diversity, and higher example difficulty of the dataset. We also find that models that display effective robustness are able to correctly classify 10% of the examples that no other current testbed model gets correct. Finally, we discuss several strategies for scaling effective robustness to the high-accuracy regime to improve the out-of-distribution accuracy of state-of-the-art models.

## 1 INTRODUCTION

The ability to generalize to data not seen during training is essential for the widespread trust and adoption of machine learning models. In practical applications of machine learning, we typically train and fine-tune on a dataset which are in-distribution (ID) with each other, but when deployed the model will face shifts from this distribution, and empirically the vast majority of models show a significant drop in performance from ID data to out-of-distribution (OOD) data (Quiñonero-Candela et al., 2009; Recht et al., 2019; Biggio & Roli, 2018; Szegedy et al., 2013; Hendrycks & Dietterich, 2019; Azulay & Weiss, 2018; Shankar et al., 2019; Gu et al., 2019; Torralba & Efros, 2011). Common examples include data captured in a different environment, like time of day or geographical location (Koh et al., 2020); noise or small corruptions of the input data (Hendrycks & Dietterich, 2019; Geirhos et al., 2018); or adversarial examples created to explicitly fool neural networks into making incorrect predictions (Szegedy et al., 2013; Biggio & Roli, 2018).

Although the performance gap on OOD shifts is pervasive, it follows an intriguing pattern: *there is a clear linear relationship between a model's final performance on ID and OOD data* (Taori et al., 2020; Recht et al., 2019; Yadav & Bottou, 2019; Miller et al., 2020). In other words, given a model's performance on an ID test set, a linear fit can accurately predict what the performance drop will be on the OOD test set. This also implies that a certain amount of improvement on OOD accuracy can be explained by an improvement on ID accuracy. The linear relationship holds across a wide range of models and is well-established for several robustness benchmarks, including image classification on both synthetic and natural distribution shifts (Recht et al., 2018; Yadav & Bottou, 2019; Recht et al., 2019; Taori et al., 2020), 2D object detection (Shankar et al., 2019), and question-answer models in NLP (Miller et al., 2020). Since even the highest-performing models will still have a gap between ID and OOD accuracy, the linear relationship reveals that our current methods are insufficient for addressing OOD shift.

Models which lie above the linear fit are said to exhibit *effective robustness* (ER) (Taori et al., 2020), which measures the model's OOD accuracy relative to the fit (see Figure 1(a) for an illustration). Models with nonzero ER deviate in their OOD behavior in a manner that is qualitatively and

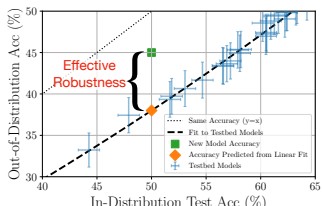

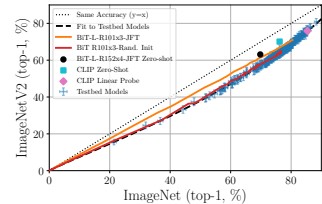

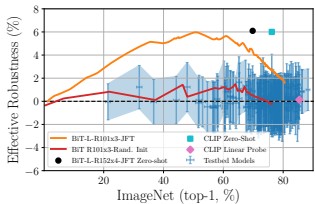

(a) Illustrating the definition of effective robustness (ER)

(b) Accuracies on ImageNet and ImageNetV2

(c) ImageNet accuracies vs. ER on ImageNetV2

Figure 1: *During fine-tuning on ImageNet, pre-trained models exhibit effective robustness (ER) while randomly-initialized models do not.* However, the ER of pre-trained models diminishes throughout the fine-tuning process and vanishes altogether at the end of training. (a) illustrates the definition of ER that we define more precisely in Section 3. Subfigure (b) and (c) show ImageNet test accuracy on the x-axis for either randomly initialized or pre-trained models throughout ImageNet training (we evaluate accuracy on checkpoints every epoch of training). (b) plots the ImageNet-V2 test accuracy on the y-axis while (c) shows the effective robustness for the same models. ER can also be achieved by zero-shot models. In (b) and (c) we also show a BiT model that was trained on JFT and evaluated on ImageNet using a class-map between the two sets of targets (see Appendix C.7), and we show the best performing zero-shot and linear probe CLIP models from Radford et al. (2021a). The BiT and CLIP zero-shot models have comparable ER, but at different ID accuracy, while the linear probe CLIP model has no ER. Standard testbed models (see Section 4) fully trained on ImageNet are in blue and are shown with 95% Clopper-Pearson confidence intervals.

quantitatively different from what we can currently achieve. We seek to thoroughly investigate when the pre-training and fine-tuning paradigm gives rise to models with nonzero ER, and understand factors that control this behaviour.

Models with high ER ($> 1\%$) are exceedingly rare. First, the most notable example is the recently proposed zero-shot CLIP model (Radford et al., 2021b), and subsequent work (Wortsman et al., 2021) has found that weight-space ensembling of zero-shot and fine-tuned CLIP-style models can improve both ID and OOD accuracy and produce models with high ER. Second, only a handful of the 204 ImageNet models evaluated in Taori et al. (2020) were found to have non-zero ER. Though most of these effectively robust models identified so far were pre-trained on large datasets, the majority of models pre-trained this way exhibit no ER, and we currently do not know when additional data helps; what effect the choice of architecture has; as well as the effects of dataset and distribution shift on these findings.

In this work, we present an empirical study of the evolution of ER throughout fine-tuning. By studying the evolution, we find intriguing properties that are missed by focusing only on models at convergence: pre-trained models exhibit ER, while randomly-initialized models do not (see Figure 1).

**Summary of Contributions:**

- (Section 5.1, 5.3) We identify pre-trained, fine-tuned models as an entire class of effectively robust models (that match the ER of CLIP (Radford et al., 2021a)) and investigate how details such as model size, dataset size, and example difficulty influence ER. We find that the vanishing of ER at convergence depends on the distribution shift, and on some datasets pre-trained, fine-tuned models may still exhibit non-zero ER at convergence.

- (Section 5.4) We analyze properties of pre-trained models at the peak of their ER, including particular metrics defined in previous theoretical work (Mania & Sra, 2020). We find that effectively robust models make remarkably dissimilar predictions compared to standard models, and are able to correctly classify 10% of the examples that no other model gets correct.

- (Section 5.5) We find that pre-trained models gradually lose their ER throughout fine-tuning, even as both the ID and OOD accuracies of the model simultaneously increase. We discuss several potential solutions to mitigate this problem, but find that none of them are able to maintain high ER at high ID accuracy.

## 2    RELATED WORK

Here we review some prior findings that are key for understanding the context of our results.

**Linear trends under distribution shift.** In recent replication efforts, researchers carefully recreated new test sets for the CIFAR-10 and ImageNet classification benchmarks (Recht et al., 2018; 2019). Despite following the original collection procedures as closely as possible, some shift was introduced. Evaluating an extensive testbed of trained models on the original and new tests revealed a clear relationship between accuracy on the original and new tests that is well-captured by a linear fit with positive slope. Follow-up work also found linear trends for new test sets on MNIST (Yadav & Bottou, 2019) and the SQUAD question-answer dataset (Miller et al., 2020), geographic distribution shifts for 3D Object Detection in self-driving cars (Sun et al., 2019), and synthetic distribution shifts (Taori et al., 2020). We find that models pre-trained on larger and more diverse datasets can break the linear trend of accuracy under distribution shift in the middle of the fine-tuning process, and we investigate factors that affect this.

**Examples of effectively robust models**. Models that lie off the linear trend (*effectively robust* models) are historically extremely rare. In a recent extensive empirical study, researchers evaluated 204 trained ImageNet models spanning a wide range of architectures and training techniques on several popular natural and synthetic robustness benchmarks for ImageNet (Taori et al., 2020). Across several natural robustness benchmarks, the main outliers with positive ER and high accuracy on ImageNet were all models that were pre-trained on larger and more diverse data than the ImageNet training set. They included a ResNet152 model trained on 11,000 ImageNet classes (Wu, 2016), several ResNeXt models trained on 1 billion images from Instagram (Mahajan et al., 2018), and the EfficientNet-L2 (NoisyStudent) model trained on a Google-internal JFT-300M dataset of 300 million images (Xie et al., 2020). However, not all models that were pre-trained on larger datasets showed effective robustness.

Another recent work, (Radford et al., 2021a), observed that zero-shot CLIP classifiers have larger ER than CLIP models that had been fine-tuned to the downstream task. While the CLIP model was pre-trained with a contrastive loss that combined components from natural language processing and image classification, our results show that pre-trained ImageNet models that are evaluated in a zero-shot manner also have high ER, and, similar to the CLIP model, these image classification models lose their ER once they are fine-tuned to the downstream task. Since the completion of this paper, subsequent work (Wortsman et al., 2021) has proposed weight-space ensembles for fine-tuning (WiSE-FT) that interpolates between zero-shot and fine-tuned weights to find models with high ER as well as high ID and OOD accuracy. This method is currently limited to CLIP-style models, and in this work we mostly focus on traditional supervised image classification models.

**Model similarity.** Recent work offers a theoretical explanation, relying on an assumption of *model similarity* (Mania & Sra, 2020), for why classifier accuracies follow a linear trend under distribution shift. Empirically, Mania et al. (2019) observed that the labeling assignments across a wide range of ImageNet models are significantly more similar to each other than would be expected by chance, which suggests that the size of the class of function approximators that neural networks learn is smaller than what may be expected a priori. Under the assumption of model similarity, Mania et al. (2019) proves that models' accuracies, when evaluated on two distributions, must be approximately collinear, unless the size of the distribution shift is large in a certain sense. In our work, we identify and analyze models that are not collinear with other models when evaluated on two distributions, and we find that such models break the assumption of model similarity sufficient to prove the existence of the linear fit.

## 3    EFFECTIVE ROBUSTNESS

The linear fit to testbed model accuracies on ID and OOD data provides a baseline for how improvements in OOD accuracy are tied to improvements in ID accuracy based on current methods. Effective robustness (ER), a metric for robustness proposed in Taori et al. (2020), measures the difference between a model's OOD accuracy and that predicted from the baseline. (See Figure 1(a) for an illustration.)

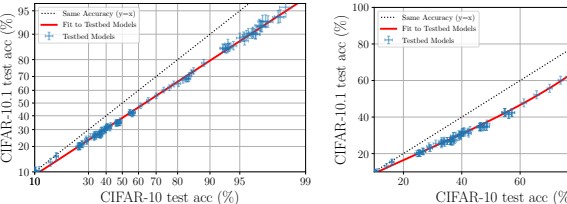

Figure 2: **Testbed model accuracies ( Recht et al. (2018)) show a linear relationship in a rescaled logit-space Taori et al. (2020).** (a) shows the linear fit in logit-space, and (b) shows the same fit transformed back to linear-space.

To define ER, we fix an ID and OOD test set and let $\beta(x)$ be the baseline predicted OOD test set accuracy for a given accuracy $x$ on the ID test set. Given a set of models evaluated on both the ID and OOD test sets, $\beta$ can be computed by performing a log-linear fit between the models' accuracies on the two test sets. Similar to Taori et al. (2020), we empirically find that the best linear fit comes from rescaling the accuracies using the logit function, $\text{logit}(\alpha) = \log\left(\frac{\alpha}{1-\alpha}\right)$, before performing the linear fit. In Figure 2 we show the log-linear fit for CIFAR-10 vs. CIFAR-10.1 in logit-space as well as transformed back to linear-space. See Appendix A for more details.

The *effective robustness* of a model $f$ is then defined as

$$\rho(f) = \text{acc}_{\text{out}}(f) - \beta(\text{acc}_{\text{in}}(f)), \tag{1}$$

where $\text{acc}_{\text{in}}(f)$ and $\text{acc}_{\text{out}}(f)$ represent the model's accuracies on the ID and OOD test sets, respectively. Graphically, ER represents the distance above the testbed line to a model's OOD accuracy, as shown in Figure 1(a).

We note that ER is distinct from the absolute accuracy a model can achieve on the OOD data. For instance, the effectively robust model (green point) in Figure 1(a) still has low absolute OOD accuracy relative to other models. An ideal model achieves both high ER and high absolute OOD accuracy.

## 4 EXPERIMENTAL SETUP

In this section, we discuss which fine-tuning datasets, pre-trained models, robustness benchmarks, and linear fits we use in our experimental evaluation.

**Fine-tuning datasets.** We evaluate ER throughout fine-tuning (see Appendix B.2 for our definition of fine-tuning) on both ImageNet and CIFAR-10 since researchers commonly fine-tune pre-trained models to these popular benchmarks. For CIFAR-10 in particular, there are several widely available pre-trained ImageNet models that we can easily transfer to the CIFAR-10 dataset.

**Pre-trained models.** The majority of the pre-trained models we study are Big Transfer (BiT) models (Kolesnikov et al., 2020), which use a ResNet-v2 architecture (He et al., 2016b), with Group Normalization (Wu & He, 2018) and Weight Standardization (Qiao et al., 2019). The BiT models come in three different flavors: BiT-S, BiT-M and BiT-L, where S, M and L indicate if the pre-training was done on ILSVRC-2012 (which we denote as ImageNet-1k or just ImageNet) (Russakovsky et al., 2015), ImageNet-21k (Deng et al., 2009), or JFT (Sun et al., 2017), respectively. In addition, these models may be further fine-tuned, and "1k", "21k" or "JFT" indicate the dataset it was fine-tuned on. For example, BiT-M-R152x4-1k is the ResNet-152x4 model that was pre-trained on ImageNet-21k ("M") and fine-tuned on ImageNet-1k ("1k"). We also study a series of BiT models that were pre-trained in Djolonga et al. (2021) on different amounts of data from JFT.

Table 1: The range of experiments we run are represented with the cross product of the models, OOD dataset and pre-training dataset listed below.

| Model | OOD | Pre-training dataset |
|---|---|---|
| AlexNet, ResNet-18,-50, -152, VGG11_BN, Wide-ResNet-50-2 | CIFAR-10.1 | ImageNet-1k |
| BiT-R{50x1,101x3,152x4} | CIFAR-10.1 ImageNetV2 | ImageNet-1k ImageNet-21k, JFT |
| BiT-R101x3 | ObjectNet, ImageNet-R | JFT |

We also study six ImageNet pre-trained PYTORCH (Paszke et al., 2019) models: AlexNet (Krizhevsky, 2014), ResNet-18, -50 and -152 (He et al., 2016a), VGG-11 with Batch Normalization (Simonyan & Zisserman, 2014), and WideResNet-50-2 (Zagoruyko & Komodakis, 2016).

**Robustness benchmarks.** Both CIFAR-10 and ImageNet have several well-established robustness benchmarks. We choose to focus on naturally occurring distribution shifts, rather than synthetic shifts which modify images, because they are more realistic and because there are several robustness interventions that work well on synthetic shifts but do not transfer to natural distribution shifts Taori et al. (2020). Thus, for CIFAR-10, we evaluate robustness on CIFAR-10.1, which was created in the replication study of Recht et al. (2018) and is one of the few natural distribution shift benchmarks available for it; and for ImageNet, we use the ImageNetV2 (Recht et al., 2019), ObjectNet (Barbu et al., 2019), and ImageNet-R (Hendrycks et al., 2020) test sets.

**Linear fits.** Measuring ER requires establishing a linear relationship between a model's accuracy on ID and OOD data. To define the linear relationship between the original test accuracy and the OOD test accuracy on CIFAR-10 and ImageNet, we require a testbed of models that span the range from low to high accuracy. For ImageNet, we use the publicly available results from Recht et al. (2019), and for CIFAR-10 we use the results from Recht et al. (2018).[1]

## 5 RESULTS

We overview our main experimental results. First, we establish that pre-trained models exhibit ER during fine-tuning, and we examine how properties of the pre-training setup, such as model size, dataset size, and example difficulty, influence the magnitude of the ER. Finally, we explore ideas that attempt to maintain high ER at the end of fine-tuning.

### 5.1 PRE-TRAINED MODELS HAVE HIGH EFFECTIVE ROBUSTNESS

We find that models pre-trained on large and diverse datasets exhibit high effective robustness during fine-tuning, while randomly initialized models do not at any point during training (Figure 3). As described below, this observation is invariant to the choice of robustness benchmark (CIFAR-10.1, ImageNet-V2, ObjectNet, and ImageNet-R); model architecture; and the details of the fine-tuning, such as learning rate, optimization algorithm, type of fine-tuning (last-layer versus whole model), and loss function.

**CIFAR-10.** We evaluate several architectures, including AlexNet, ResNet-18, ResNet-50, ResNet-152 and VGG-11-BN, throughout fine-tuning on CIFAR-10 using both ImageNet pre-trained initialization and random initialization. At every epoch of training, we measure ER using CIFAR-10.1 as the OOD test set and CIFAR-10 as the ID test set. Figure 3(a) plots the ER (%) versus CIFAR-10 test accuracy (%) throughout fine-tuning for all of the ImageNet pre-trained architectures, as well as the randomly initialized architecture that achieved the maximum ER (for the full comparison see Appendix 10). We see that all of the ImageNet pre-trained models have significantly higher maximum ER than either the testbed models or the best performing randomly initialized model. Moreover, we observe that ER generally increases throughout fine-tuning, peaks, and then gradually disappears towards the end of fine-tuning.

**ImageNet.** For models fine-tuned on ImageNet, we measure ER for JFT pre-trained models at every epoch of training using either ImageNetV2, ObjectNet, or ImageNet-R as the OOD test set and ImageNet as the ID test set. We report ImageNet top-1 accuracy, and for ImageNet-R and ObjectNet, we compute accuracy over the respective subset of classes for the two benchmarks. Figure 3(b) shows the ImageNetV2 ER, and Figures 3(c) and 3(d) shows ER on ImageNet-R and ObjectNet respectively. In all cases, the pre-trained model reaches a maximum ER during fine-tuning which then decreases as we continue fine tuning.

For the ObjectNet and ImageNet-R evaluation, we fine-tune a BiT-L-R152x4-JFT model on ImageNet and evaluate the *same model* on ImageNetV2, ObjectNet, and ImageNet-R. Though the model's ImageNet accuracy is the same in all cases, for ImageNetV2, the ER vanishes (OOD accuracy drops to the baseline set by the testbed models), while for ImageNet-R and ObjectNet there is still a non-zero

---

[1]Additionally, we add several low-accuracy scikit-learn models which we received in personal communication and which we expect to be publicly available soon.

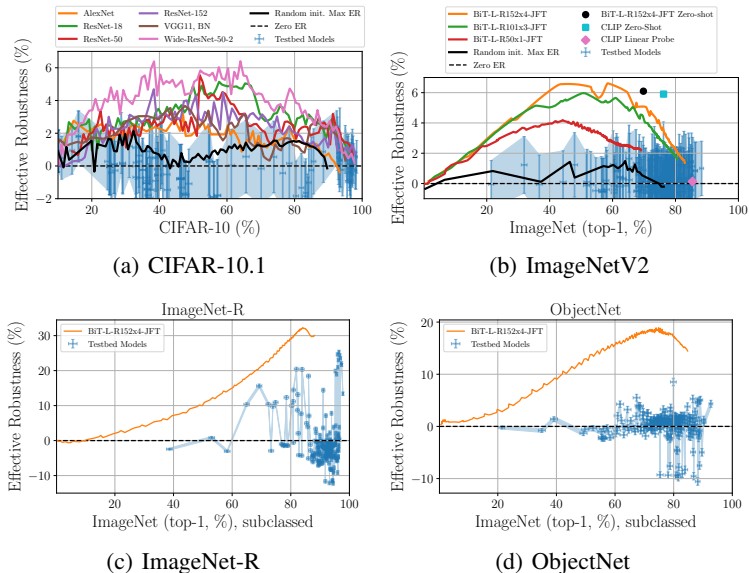

(a) CIFAR-10.1             (b) ImageNetV2

(c) ImageNet-R             (d) ObjectNet

Figure 3: **Pre-trained models consistently have high ER across different choices of architectures on both ImageNet and CIFAR-10.** We plot ER (%) versus ID accuracy (%) at every epoch of training as we fine tune (a) pre-trained ImageNet models on CIFAR-10 or (b,c,d) pre-trained JFT models on ImageNet. Figure (a) shows the ER on CIFAR-10.1 while Figures (b,c,d) show the ER on ImageNetV2, ImageNet-R, and ObjectNet, respectively. For each plot, we also show the ER and accuracy achieved by converged testbed models in blue with 95% Clopper Pearson confidence intervals. For (a) CIFAR-10.1 and (b) ImageNetV2, we trained a randomly initialized version of all the model architectures and we report the model with the highest maximum ER during training (solid black curve). We observe that the pre-trained models all exhibit high ER in the middle of fine-tuning, which eventually decreases as fine-tuning proceeds. In contrast, the randomly initialized models do not have significant ER relative to the testbed models.

ER at the end of training. This observation is consistent with Taori et al. (2020), which found that converged pre-trained models are more likely to exhibit nonzero ER on ObjectNet or ImageNet-R than ImageNetV2.

**Model architecture.** We find that pre-trained models with larger architectures exhibit higher ER. For ImageNetV2 in Figure 3(b), we evaluate multiple BiT ResNets with increasing depth and width (R150x1, R101x3, R152x4) that were pre-trained on JFT, and we find that models with more parameters have a higher peak in ER. Similarly, for CIFAR-10.1, Figure 3(a) shows that larger, more recent architectures such as the Wide-ResNet-50-2 exhibit higher effective robustness than smaller architectures like AlexNet. Moreover, Figure 4(b) shows that max ER on CIFAR-10.1 increases for BiT models of increasing depth and width as we vary the pre-training dataset.

**Details of training.** We performed a detailed study of a variety of components of the optimization process, and we found that the ER is not sensitive to these details. See Appendix C for variations in learning rate, loss function, type of optimization (full model vs. last layer).

## 5.2 ZERO-SHOT EVALUATION

Recent work (Radford et al., 2021a) introduced the CLIP architecture that allowed for zero-shot evaluation on image classification datasets like ImageNet and the ImageNet robustness benchmarks. The zero-shot CLIP models has high amounts of ER, but it is unclear if the ER comes from the new and very large dataset used for training, the natural language model component, the contrastive training objective, or its zero-shot evaluation. In Figure 3(b) we show the ER of the best zero-shot and linear probe CLIP models from Radford et al. (2021a). We also show results from zero-shot evaluation on a BiT model that was pre-trained on JFT. This is done by creating a mapping between similar classes of JFT and ImageNet. While the BiT model has about 5% lower ImageNet top-1 accuracy, it matches the amount of ER that the CLIP model has. This is surprising, and suggests that the zero-shot component of CLIP plays a significant role in the high value of ER CLIP achieves.

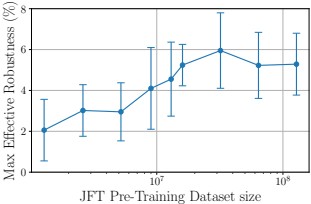

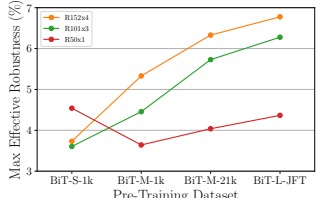

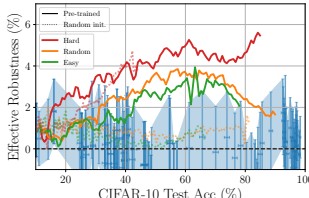

(a) Max ER
vs. fraction of JFT dataset size

(b) Max ER vs.
different pre-training datasets

(c) Varying the difficulty of the
fine-tuning dataset

Figure 4: **The maximum ER depends both on the pre-training and the fine-tuning dataset.** (a) We evaluate BiT-R101x3 models pre-trained on varying fractions of the JFT dataset throughout fine-tuning on the CIFAR-10 dataset, and we find that maximum CIFAR-10.1 ER increases with the size of the pre-training dataset. Error bars represent standard deviation over 5 runs. (b) We evaluate several BiT models of increasing size (R50x1, R101x3, R152x4) that were pretrained on either ImageNet (S), ImageNet-21K (M), or JFT (L) throughout fine-tuning on CIFAR-10. (The BiT-M-1K was first pre-trained on ImageNet-21K and then ImageNet-1k). Maximum CIFAR-10.1 ER increases as the model size and pre-training dataset size and diversity increase. (c) The maximum CIFAR-10.1 ER also depends on the difficulty of the fine-tuning dataset. We select examples based on the C-score Jiang et al. (2020) of CIFAR-10 and train on either the 5000 easiest, hardest or random images in the training set. Training on harder examples yields more ER for both pre-trained and randomly initialized models, but lower CIFAR-10 test accuracy than training on random examples.

## 5.3 THE IMPACT OF DATA ON EFFECTIVE ROBUSTNESS

We find that the maximum ER depends on both the pre-training dataset and the data the model is fine-tuned on. As we discuss below, larger and more diverse pre-training datasets, as well as more difficult fine-tuning examples, all increase the magnitude of the ER peak.

**Dataset size and diversity.** In Figure 4(a), we pre-train models on randomly sampled fractions of JFT. We observe that maximum ER increases as the amount of pre-training data increases but eventually plateaus for the largest dataset sizes. We suspect the reason for the plateau is the fixed number of iterations used across those experiments, which eventually serves a bottleneck for performance improvements as reported by prior work (Hernandez et al., 2021; Nakkiran et al., 2020). Figure 4(b) shows that maximum ER increases as we pre-train on larger and more diverse datasets.

We also find that pre-training first on a larger dataset and then switching to a smaller dataset leads to lower ER during the final fine-tuning task than a setup that only pre-trains on the larger dataset. To see this, consider the BiT-M-1k and BiT-M-21k in Figure 4(b). Both models were initially pre-trained on ImageNet-21k (as denoted by the 'M'), but the BiT-M-1k model was additionally fine-tuned on ImageNet-1k as part of the pre-training procedure. We then evaluated the ER of both models as we fine-tuned them to CIFAR-10. Figure 4(b) shows that the extra fine-tuning step on ImageNet-1k gives about a 1% drop in ER relative to the BiT-M-21k model.

**Example difficulty.** The ER during training also depends on the data the model is being fine-tuned on. In Figure 4(c) we select the 5000 easiest, hardest or random examples from CIFAR-10 training set and only fine-tune on these examples. The difficulty of each example is measured by the C-score (Jiang et al., 2020). Training on easier (harder) examples gives less (more) ER, but training on only the hardest or easiest examples gives lower final accuracy than training on the examples of random difficulty. See Appendix C.9 for more details.

## 5.4 ANALYSIS AT MAXIMUM EFFECTIVE ROBUSTNESS

To understand the differences between models that exhibit ER and testbed models, we compare the predictions from the former – both at the peak ER and at the end of training – and the latter.

**Dominance probability.** Recent work (Mania & Sra, 2020) showed that the linear trend between the ID and OOD accuracies will arise if two assumptions are satisfied: low dominance probability and a specific notion of distributional closeness. Dominance probability (Mania & Sra, 2020) is a similarity

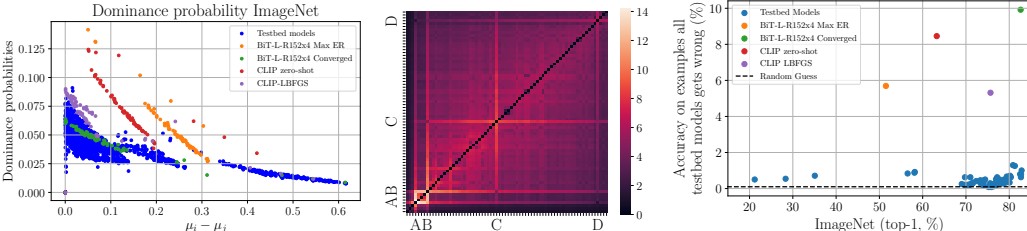

(a) Dominance prob. distribution     (b) Pairwise dominance prob. (c) Difference from testbed predictions

Figure 5: **Models with high effective robustness have higher dominance probabilities than models with zero effective robustness on ImageNet.** We analyze the predictions of the following four models: BiT-L-R152x4 Max ER (A), CLIP zero-shot (B), CLIP-LBFGS (C), and BiT-L-R152x4 Converged (D). (a) For each pair of models in the testbed plus these four models, we plot their dominance probability versus their accuracy difference. We find that the CLIP zero-shot model and the pre-trained BiT-L-R152x4 model have higher dominance probabilities on ImageNet than any of the models in the testbed, but once the CLIP and BiT-L models are fine-tuned on ImageNet, they align more with the dominance probability distribution of the testbed models. (b) We show the pairwise comparison of the dominance probability on ImageNet between all testbed models (not labeled) in addition to our four models. We sort models in order of increasing top-1 accuracy on ImageNet (see Appendix C.10 for a bigger version with labels for all models). We see that the four labeled models make very different predictions relative to the testbed. (c) For each of the plotted testbed models, we plot the percent of examples that none of the *other* testbed models get correct. Considering the 4.8% of all examples that none of the testbed models get right, the best BiT model pre-trained on JFT gets 10% of them correct and the zero-shot CLIP model gets approximately 8% correct.

metric between the predictions of two models, measuring what fraction of examples the model with lower overall accuracy gets correct that the model with higher overall accuracy gets wrong. (See formal definition in Appendix C.10.) In this work, we find that the dominance probability assumption does not hold for models with ER, but that the assumption of distributional closeness still holds.

In Figure 5, we see that the dominance probability of a pre-trained BiT model early-stopped at its maximum ER is shifted relative to the models in the testbed, while the pre-trained BiT model that was fine-tuned to convergence (and has low ER) has dominance probability similar to the models in the testbed. We also find that the dominance probability for the pre-trained BiT model is very similar to that of a zero-shot CLIP model (see Appendix B.3 details on fine-tuning). Even though the fine-tuned CLIP model has close to zero ER (see Figure 3(b)), we see in Figure 5(a) and 5(b) that its dominance probability is also higher than that of standard testbed models. Unlike the BiT JFT pre-trained model, the CLIP model was pre-trained with contrastive learning and text embeddings, and we hypothesize that these elements may contribute to this effect.

**Examples that only effectively robust models get right.** Models that have high ER are able to correctly classify images that no other testbed model gets correct. We select the 4.8% of images that none of the testbed models get correct, and we find that the ER model with the best ID performance gets 10% of these examples correct, i.e. 237 examples, divided among 198 different classes, with a maximum number of three examples in each class. After manual inspection of all examples, we do not see any pattern among the 237 images.

**Remarks.** Since testbed models all have relatively low pairwise dominance probabilities, there are a set of easy examples correctly classified by all models, but harder examples are only classified correctly by increasingly higher accuracy models. This suggests that images can be placed on a scale of difficulty, and that this notion of difficulty is shared by all the testbed models. In contrast, the high dominance probability of the effectively robust models suggests that they have a different notion of difficulty compared to that of the testbed models, i.e. some images that are difficult for testbed models are not as difficult for effectively robust models. As further evidence, effectively robust models are able to classify some images correctly that no other testbed model gets correct. We hypothesize that the different notion of difficulty of the effectively robust models comes from pre-training on larger and more diverse data, which could expose them to examples that are dissimilar from those in the fine-tuning dataset.

### 5.5 Maintaining Effective Robustness at High Accuracy

In the previous sections, we have seen that models pre-trained on larger and more diverse datasets exhibit robustness properties during fine-tuning, but that, depending on the choice of OOD dataset, the ER can vanish as the model converges towards the end of training. In this section we explore several ideas for ways to maintain the high ER as the model reaches high accuracies, but ultimately we find that none of the techniques are able to avoid the loss of ER.

**Replay buffer.** Deep neural networks that are subject to training on two distinct tasks sequentially commonly forget learned information about the first task after training on the second, a phenomenon known as catastrophic forgetting (Kirkpatrick et al., 2017; Toneva et al., 2019). A possible hypothesis, therefore, is that retaining learned information about the pre-training dataset – for instance, by maintaining high accuracy on this dataset during the course of fine-tuning to downstream tasks – may have a beneficial effect on ER. We explore this possibility by implementing a popular technique for mitigating catastrophic forgetting – the use of a "replay buffer" (Ratcliff, 1990; Rolnick et al., 2019).

Without a replay buffer, we find that a ResNet-18 model pre-trained on ImageNet and fine-tuned on CIFAR-10 has a drop in ImageNet test accuracy from 69% to 39% when early stopped at 92% CIFAR-10 test accuracy. When simultaneously fine-tuning on CIFAR-10 with a replay buffer, we retain 63% ImageNet test accuracy, but we observe no improvement in retained ER. See Appendix C.11 for more experimental details and results.

**Mapping between different class sets.** Using the same constructed map of classes used for zero-shot evaluation (c.f. Section 5.2 and Appendix C.7), we can use the target classes of the pre-training dataset for fine-tuning. We use the mapping from CIFAR-10 to ImageNet classes, and we use this for fine-tuning on CIFAR-10 initializing with the pre-trained prediction head.

First, we explored fine tuning on just CIFAR-10 images with the 1000 ImageNet target classes. Second, we used a replay buffer for combined fine-tuning on CIFAR-10 and ImageNet using the same prediction head. In both setups we see no significant change in ER at the end of fine-tuning. See Appendix C.12 for further details on the experiments and results.

**Example difficulty.** In Section 5.3 we saw that training on harder examples during fine-tuning increases the ER, but prohibits reaching high final accuracy. We study different training schedules where we vary the difficulty of examples during training, but we find that none of the proposed methods are sufficient to maintain high ER and reach high test accuracy at the end of training.

## 6 Conclusion

Machine learning models currently experience an undesirable yet predictable drop in accuracy on OOD shifts. The exceptions from this trend – models with ER – experience less of a drop in OOD accuracy compared to standard testbed models at the same ID accuracy. The ultimate goal is to produce ER models at state-of-the-art accuracies; however, none of the known models with high ER are in the high-accuracy regime, and the models with the highest OOD accuracy have no ER. One way to approach this problem is to understand why some models have ER so that we can replicate this behavior for state-of-the-art models and decrease the OOD performance gap.

In this work, we demonstrate that pre-trained models in the middle of fine-tuning, as well as zero-shot pre-trained models, represent an entire class of models that exhibit high amounts of ER. With this rich set of effectively robust models, we take some initial steps towards understanding what is unique about them that causes them to outperform testbed models on OOD shifts. Our results suggest that properties of the pre-training or fine-tuning datasets, such as the size, diversity, or difficulty, cause effectively robust models to learn an entirely different difficulty scale than that of standard testbed models. Moreover, because ER models make different predictions than standard models and are able to correctly classify examples that no standard models get right, these models are valuable for applications such as model ensembling, which requires prediction diversity.

From our work, we know that pre-training provides a clear benefit to ER, but that fine-tuning on the ID task reduces ER. The future challenge lies in constructing mechanisms to enhance ER in the high accuracy regime. We hope our initial implementations of and investigations into such techniques will be a launching point for researchers for further experimentation.

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

## A    SCALING FOR LINEAR FIT

When modeling the linear relationship between the original and new test set for CIFAR-10 and ImageNet, applying a nonlinear scaling to the accuracy $\alpha$ has shown to improve the linear fit, especially when considering both very high and low performing models. Recht et al. (2018; 2019) applies a probit scaling to the accuracies, and Taori et al. (2020) also considers a logit scaling.

Probit scaling uses the probit function

$$\text{probit}(\alpha) \equiv \Phi^{-1}(\alpha), \tag{2}$$

where $\Phi^{-1}(\alpha)$ is the inverse of the cumulative distribution function of the standard normal distribution; the logit scaling uses the logit function

$$\text{logit}(\alpha) \equiv \log\left(\frac{\alpha}{1-\alpha}\right), \tag{3}$$

We compare the goodness of the fit for each of the scalings in Table 2. We conclude that the logit scaling gives the best linear fit for both CIFAR-10 and ImageNet and will use this scaling throughout this work. In Figure 2 we show the linear fit in a logit-scaled space and transformed back to linear space for CIFAR-10.

Table 2: $R^2$ values for linear fit for different scaling functions

| SCALING | CIFAR-10 | IMAGENET |
|---------|----------|----------|
| LINEAR  | 0.981    | 0.984    |
| PROBIT  | 0.996    | 0.995    |
| LOGIT   | **0.998** | **0.996** |

**Numerical fit values.** All linear fits throughout this work were computed by first transforming to logit space (see Section A) and then fitting using `scipy.stats.linregress`. Eq. 4 gives the functional form for our fit, and the numerical values are listed in Table 3.

$$\text{(New Acc)} = \text{logit}^{-1}\left[A \cdot \text{logit(Orig. Acc)} + B\right] \tag{4}$$

Table 3: Numerical fit values for Eq. 4

|  | $A$ | $B$ |
|---|-----|-----|
| CIFAR-10/10.1 | 0.8318 | -0.4736 |
| IMAGENET/IMAGENETV2 | 0.9225 | -0.4896 |

## B    EXPERIMENTAL SETUP

### B.1    PRE-TRAINED IMAGE CLASSIFICATION MODELS

We conduct our experiments using a diverse set of model architectures and sizes to make sure the observed phenomenon are general and not tied to any particular choice. We study two groups of models.

**ImageNet pre-trained.** The first group consists of AlexNet Krizhevsky (2014), ResNet-18, ResNet-50, ResNet-152 He et al. (2016a), VGG11 BN Simonyan & Zisserman (2014), and WideResNet-50-2 Zagoruyko & Komodakis (2016), where we use the PYTORCH Paszke et al. (2019) implementation and weights coming from `torchvision.models` v0.8.2 where the models were pre-trained on ImageNet. When fine-tuning pre-trained ImageNet models on CIFAR-10, we always replace the original classification layer with a 10-class randomly initialized classification layer.

**Big Transfer (BiT).** The second group of models consists of a series of models from Big Transfer (BiT) Kolesnikov et al. (2020). The BiT models all use a ResNet-v2 architecture He et al. (2016b),

except that they replace all Batch Normalization Ioffe & Szegedy (2015) layers with Group Normalization Wu & He (2018) and use Weight Standardization Qiao et al. (2019) in all convolutional layers. Following Kolesnikov et al. (2020), we label the ResNet architectures by their depth and the widening factor for the hidden layers, for example R50x1, R101x3 and R152x4, which are the sizes we will consider in this work.

The BiT model come in three different flavors: BiT-S, BiT-M and BiT-L, where S, M and L indicate if the pre-training was done on ILSVRC-2012 (which we denote as ImageNet-1k or just ImageNet) Russakovsky et al. (2015), ImageNet-21k Deng et al. (2009) or JFT Sun et al. (2017), respectively. In addition, these models may be further fine-tuned, and "1k", "21k" or "JFT" indicates the dataset it was fine-tuned on. For example, BiT-M-R152x4-1k is the ResNet-152x4 model that was pre-trained on ImageNet-21k ("M") and fine-tuned on ImageNet-1k ("1k").

We also consider a series of BiT models that were pre-trained by Djolonga et al. (2021) on different random fractions of the JFT dataset with no additional fine-tuning in Section 5.3.

## B.2   FINE-TUNING

Fine-tuning refers to a training setup where the weights learned by training a model on one data distribution are reused when training the model on a second data distribution. Fine–tuning can either be end-to-end, where the weights learned on the first distribution are only used for initialization when training on the second distribution, or last-layer, where only the last layer weights are updated when training on the second distribution. Fine-tuning can occur in multiple rounds using multiple successive data distributions. Typically, the earlier training datasets are larger in size and class sets and more diverse.

## B.3   CLIP MODEL

Our CLIP zero-shot model uses the publicly released pre-trained weights for the ViT-B-32 model from `https://github.com/OpenAI/CLIP`. Following the methodology from Radford et al. (2021a), we fine-tune the zero-shot CLIP model using a logistic regression classifier optimized with L-BFGS, and we determine the L2 regularization strength $\lambda$ using a hyperparameter sweep on the validation sets with ten logarithmically-spaced values between $10^{-6}$ and $10^7$.

## B.4   DATA PREPROCESSING

**CIFAR-10.**  In our experiments on CIFAR-10, we are interested in testing models that were pre-trained on ImageNet-like datasets. Since the ImageNet images are typically preprocessed to 224x224 during training, we also preprocess the CIFAR-10/10.1 images by resizing from 32x32 to 224x224 pixels. To keep the experimental setup the same, we also resize the CIFAR-10/10.1 images to 224x224 pixels when using randomly initialized models.[2] All CIFAR images are normalized using the mean = $[0.4914, 0.4822, 0.4465]$ and std = $[0.2023, 0.1994, 0.2010]$.

When evaluating on a test set, or when extracting the features from the penultimate ResNet layer when training only the last layer, we do not apply any preprocessing beyond the upsampling and normalization. When training the full model on the CIFAR-10 training set (with image size 224x224), we additionally apply `RandomCrop` with padding=28 and `RandomHorizontalFlip`.

**ImageNet.**  The ImageNet training images are preprocessed with `RandomResizedCrop` to 224 pixels and `RandomHorizontalFlip`, and the test images are resized to 256 pixels and preprocessed with `CenterCrop` to 224 pixels. Both are normalized with mean=$[0.485, 0.456, 0.406]$ and std=$[0.229, 0.224, 0.225]$.

---

[2]The testbed models for CIFAR-10 and CIFAR-10.1 were using the standard 32x32 image size, but as we see in Figure 8(b) and 10(b), the ER of our randomly initialized models align with the testbed models, which indicates that the image size does not affect our results. However, we have independently confirmed that the we get the same results in our setup with image size 32x32 as in Figure 10(b).

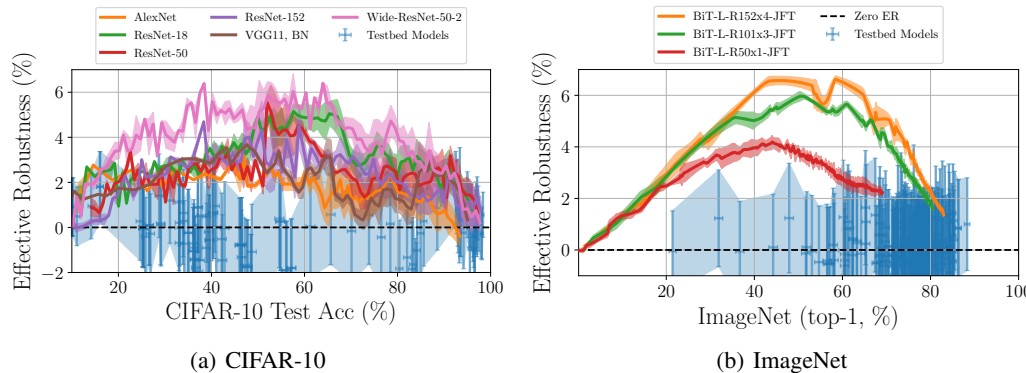

(a) CIFAR-10            (b) ImageNet

Figure 6: Multiple runs are combined by computing the ER as a binned average across the ID accuracy (using 100 bins). The shaded area around each curve shows the standard deviation per bin over 5 runs. (a) is the same data as in Figure 3(a), and (b) is the same runs Figure 3(b).

## B.5 CONFIDENCE INTERVALS, AVERAGING RUNS, AND ERROR BARS

**Testbed confidence intervals.** For the ImageNet and CIFAR testbed models, described in Section 4, we report confidence intervals on the accuracies calculated using Clopper-Pearson intervals at confidence level 95% Recht et al. (2018; 2019).

**Averaging runs.** For all measurements of ER that were averaged over multiple runs, we report the binned average value, which we calculate using `scipy.stats.binned_statistic` using 100 bins.

**Error bars.** For most of the results throughout this work, we have left out the error bars to make the plots more readable. Here, we show the typical error bars for models fine-tuned on ImageNet and CIFAR-10 in Figure 6. The reported error bars are the standard deviation from the binned average across 5 runs.

When we are measuring the maximum ER, like in Figure 4(a), we have the choice between using the standard deviation in the bin with maximum ER, or the maximum ER across all the bins. Since the number of data points in each bin can vary, we observe that the standard deviation in the maximum ER bin has more variation across different runs. So, to be conservative, we report the maximum ER across all the bins. A comparison between the two choices of standard deviation can be seen in Figure 7.

## B.6 OPTIMIZER AND HYPERPARAMETERS

Here we give the details used in the training of all our results. For all models we ran five runs and all were trained using stochastic gradient descent with momentum 0.9 and weight decay $10^{-4}$, unless otherwise specified.

The default settings to train the full PYTORCH models (see Section B.1) was using batch size 64 for 250 epochs using learning rates $[0.1, 0.01]$ for the randomly initialized model, and for 100 epochs using learning rates $[0.01, 0.001, 0.0001]$ for the pre-trained models. The main results shown in the main text and the appendix are using learning rate 0.01 and 0.001 for the randomly initialized and pretrained model, respectively, but we have checked that the general conclusions are the same across the different learning rates.

When training the BiT models, we extract the features from the penultimate layer and only train the prediction head (except for the randomly initialized BiT models where we train the full model). Unless otherwise specified, the models are trained using learning rate 0.001 with batch size 32768 for 100 epochs with no weight decay.

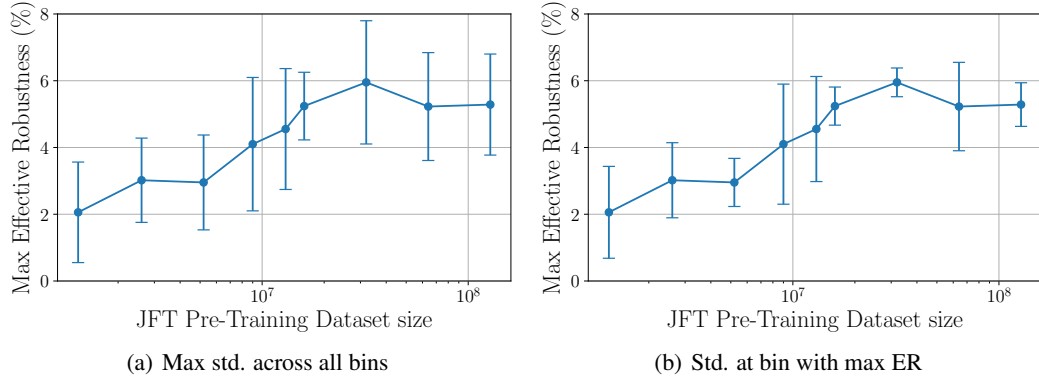

(a) Max std. across all bins

(b) Std. at bin with max ER

Figure 7: Comparison of two different ways of reporting the standard deviation (std.) for the maximum ER for the experiments in Figure 4(a). (a) shows the maximum standard deviation across all bins of CIFAR-10 ID test accuracy, and (b) uses the standard deviation in the bin that had the maximum ER. (b) has higher variation in the std. across different runs because each bin contains different number of data points, so we use (a) as a more conservative estimate as our main reported results.

## C  ADDITIONAL EXPERIMENT DETAILS AND RESULTS

This section describes we additional experimental results and details of training not described in the main paper.

### C.1  PRE-TRAINED VERSUS RANDOMLY INITIALIZED MODELS

In Section 5.1 we discussed how pre-trained models exhibit ER during fine-tuning, while randomly initialized models have no ER during training. Here we add additional experimental details that did not fit into the main paper.

**ImageNet.** In Figure 8(a) we show the ER during training on ImageNet for a BiT-L-R101x3 model that was pre-trained on JFT in comparison to a randomly initialized model of the same architecture. The pre-trained model is averaged across 5 runs and is only training the last layer. For the randomly initialized model we train the full model, but due to the high computational cost, we only perform one run. The pre-trained model is using hyper-parameters as described in Appendix B.6 while the randomly initialized model was trained following the setup as described in Kolesnikov et al. (2020).

**CIFAR-10.** In Figure 8(b) we show the exact same experiment, as described in the previous paragraph, on CIFAR-10. Again, the randomly initialized model is only run a single time due to the computational cost.

For CIFAR-10 we also perform extensive comparisons between several different architectures where we either use pre-trained or randomly initialized weights. The results are shown in Figure 10 and the hyper-parameters are as described in Appendix B.6.

### C.2  IMAGENET ROBUSTNESS BENCHMARKS

For models that are fine-tuned on ImageNet, we study the ER on ImageNetV2, ImageNet-R and ObjectNet. We train one BiT-L-R152x4_JFT model on ImageNetV2, and evaluate the same model on all three robustness benchmarks. The ER for ImageNet-R and ObjectNet was shown in Figure 3(c) and 3(d). In Figure 9 we show the same data as in Figure 3(c) and 3(d), but in term of accuracies in stead of ER. While the ER on ImageNetV2 goes to zero at the end of fine-tuning, the ER for ImageNet-R and ObjectNet remains non-zero. As discussed in Section 5.1, this is consistent with previous observations in the literature. Besides observing that the accuracy on ImageNet-R and ObjectNet stops at around 60%, we do not know why this is happening for these two datasets but not for ImageNetV2.

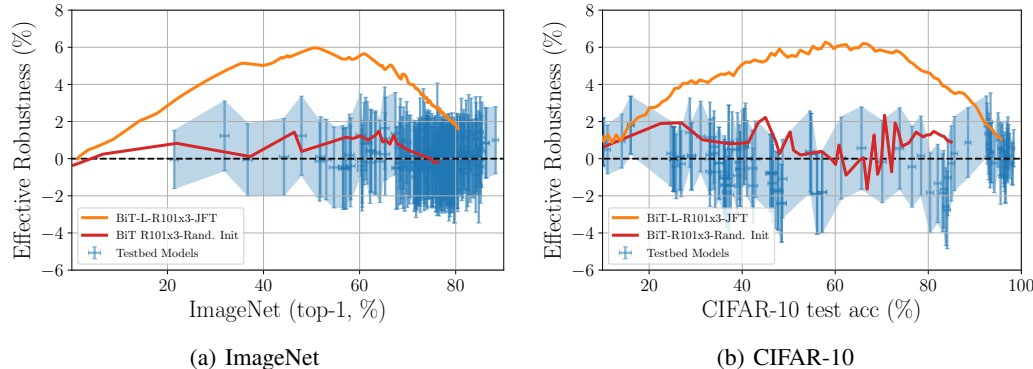

(a) ImageNet        (b) CIFAR-10

Figure 8: Comparison of the ER of a BiT-R101x3 model that is either pre-trained on JFT or randomly initialized. (a) shows the ER on ImageNetV2 when trained on ImageNet, and (b) shows the ER on CIFAR-10.1 when trained on CIFAR-10. For both datasets, the randomly initialized models show no ER while the pre-trained models do exhibit ER during training, but it vanishes as the model converges.

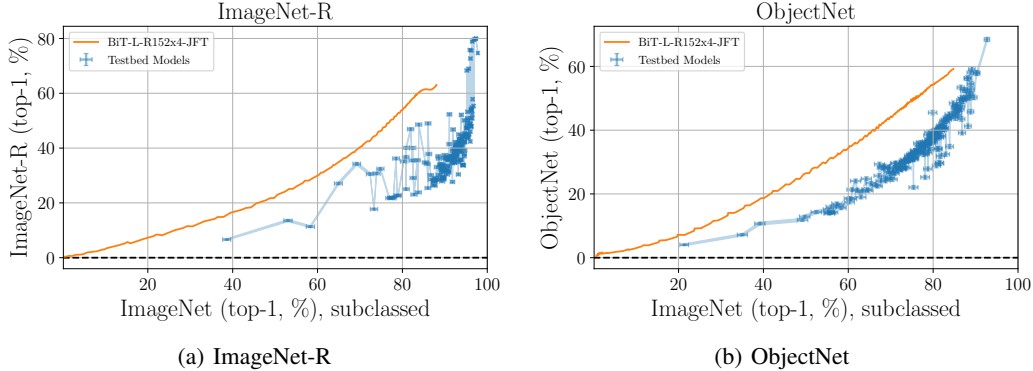

(a) ImageNet-R        (b) ObjectNet

Figure 9: Evaluation on ImageNet-R and ObjectNet for a BiT-L-R152x4_JFT model that is fine-tuned on ImageNet. Where the ImageNetV2 ER goes to zero at the end of fine-tuning, the pre-trained model still has non-zero ER.

## C.3 ARCHITECTURE INDEPENDENCE

We observe ER of pre-trained models across different model architectures in Figure 10, and none of these architectures have any significant ER when randomly initialized. We used PYTORCH models (c.f. Section B.1) pre-trained on ImageNet and replaced the prediction head with a randomly initialized layer for CIFAR-10.

Both the pre-trained and randomly initialzed models were trained with SGD with learning rates 0.01, 0.001 and 0.0001, and batch size 64, momentum 0.9, and weight-decay $5 \cdot 10^{-4}$. We show the results from averaging five runs using learning rate 0.001 in Figure 10, but the results are similar using the other learning rates (learning rate dependence is studied in Appendix C.5). The randomly initialized models were trained for 250 epochs, while the pre-trained models were trained for 100 epochs.

## C.4 LOSS FUNCTION

In this section we show that our observations about ER are not unique to models trained using cross-entropy loss, and we get the same behaviour using mean square error (MSE).

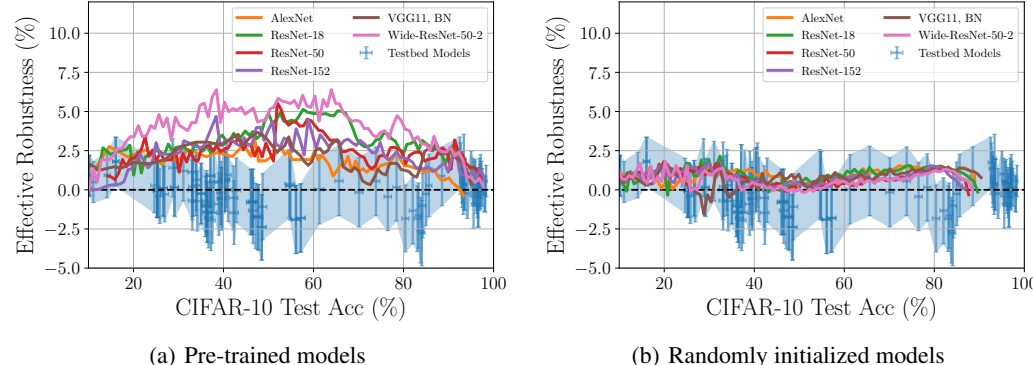

(a) Pre-trained models          (b) Randomly initialized models

Figure 10: Pre-trained models show ER for all six architectures studied, while none of the randomly initialized models do. The amount of ER does vary across different pre-trained models, but since we did not do the pre-training, we cannot control for the differences between architecture and optimization in this experiment (See Section 5.3 and Appendix C.8 and C.8 for more controlled experiments on the BiT architectures.) However, the randomly initialized models show that pre-training is crucial to get any ER at all for any of the architecture choices.

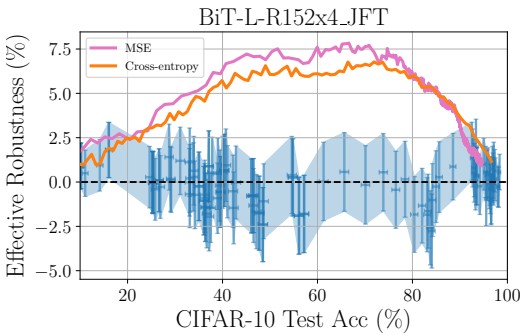

Figure 11: The rise and fall of ER occurs for different loss functions used in training. Here, we see that the ER during fine-tuning is incredibly similar between a model trained with MSE and cross-entropy loss.

For evaluation of the MSE error loss, we compute the MSE between the output logits and a one-hot target vector. We train the model using stochastic gradient descent with constant learning rate 0.001 and batch size 4096 for 100 epochs.

In Figure 11 we compare the ER duing fine-tuning of the last layer of a JFT pre-trained BiT-L-R152x4 model where one is trained using MSE and the other is trained using cross-entropy loss. In this specific experiment, the MSE trained model has higher ER at its maximum, but due to the other similarities between the evolution we do not believe this would always be the case.

## C.5 LEARNING RATE

The observed ER during fine tuning is largely independent of the learning rate used during optimization. In Figure 12 we show the ER during training on CIFAR-10 for a pre-trained and randomly initialized Wide-ResNet-50-2 using the same hyper-parameters as in Appendix C.3.

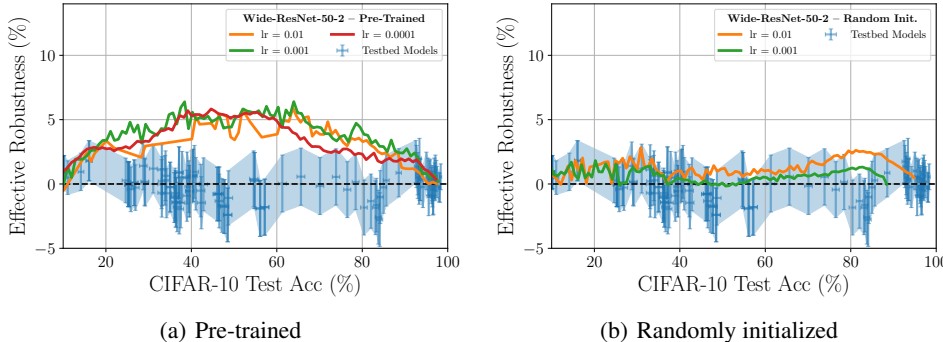

(a) Pre-trained                              (b) Randomly initialized

Figure 12: Same setup as in 10, but with varying learning rate for a fixed architecture, Wide-ResNet-50-2. The left (right) plot shows the results using pre-trained (random) weights. While there is some small variations from different settings, the conceptual difference between a pre-trained model and randomly initialized weights holds true.

### C.6  FULL MODEL VS. LAST LAYER FINE-TUNING

The ER during fine-tuning of pre-trained models is largely insensitive to whether the full network is trained or if we only train the last layer. In Figure 13 we compare the ER of full model versus last layer fine-tuning for the six PYTORCH models described in Appendix B.1.

We start both last layer and full model fine-tuning with the same pre-trained weights for that architecture. For training of the full model, we use SGD with learning rate 0.001, and batch size 64, momentum 0.9, and weight-decay $5 \cdot 10^{-4}$ for 250 epochs. When training only the last layer, we use SGD with learning rate 0.0001, and batch size 4096, and without momentum or weight-decay, and we trained for 1000 epochs. We experimented with learning rate schedulers, but saw no significant change. Despite the difference in training and hyper-parameters, we observe that both have about the same amount of ER. The biggest observed difference is that the only last-layer trained models don't reach the same final accuracy as when training the full model, which is to be expected as the last-layer model cannot change its pre-learnt features.

### C.7  ZERO-SHOT EVALUATION

Zero-shot evaluations on traditional image classifiers is made possible with hand curated maps between the classes of one dataset to another. To map from CIFAR-10 to ImageNet, we utilize the mapping from ImageNet to CIFAR-10 classes used in CINIC-10 Darlow et al. (2018), and we utilize a previously constructed map from ImageNet to JFT to evaluate the performance of BiT-L models on ImageNet in a zero-shot manner.

These maps were used for zero-shot evaluation on ImageNet and ImageNetV2 in Figure 1 and discussed in Section 5.2. The map from CIFAR-10 to ImageNet was used in Appendix C.12.

In this section we investigate different ways of combining the multiple logits corresponding to each CIFAR-10 class to make predictions using a model pre-trained on ImageNet. Given a CIFAR-10 class, we collect the corresponding ImageNet softmax scores and combine the probabilities using three different functions: max, mean and sum. We show the results in Figure 14, where we see that most of the zero-shot models have some amount of ER, but are not significantly more robust than pre-trained models in the middle of fine-tuning. We see no significant difference between the three combination functions we test, and we use the max as the default in all other plots throughout our work.

### C.8  DATASET SIZE AND DIVERSITY

In Section 5.3 we summarized our experiments where we vary the pre-training dataset for different fixed BiT architectures. Here, we will include additional experimental details and observations.

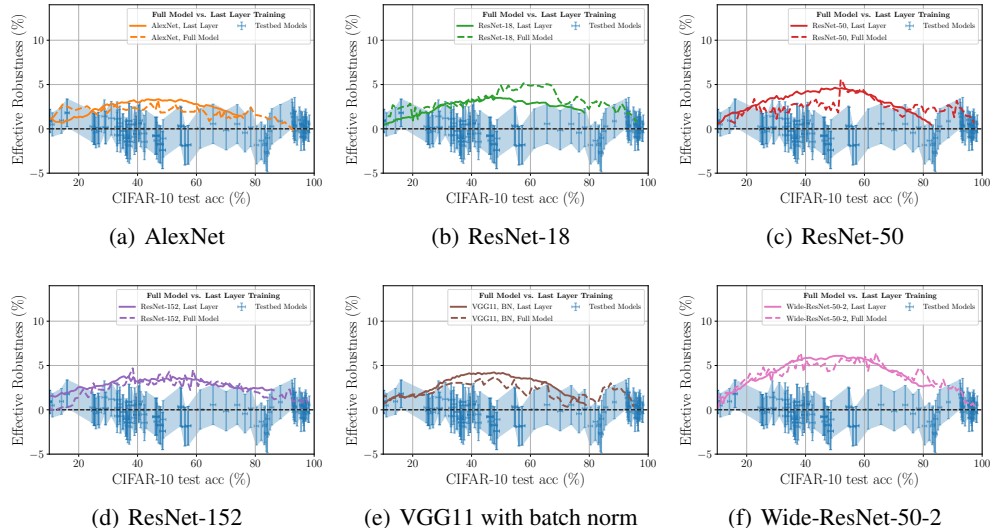

Figure 13: The ER of pre-trained models is mostly insensitive to whether the full model is trained during fine-tuning or just the last layer. We observe this across six different model architectures, and the only significant observed difference is that the full-model training reaches higher final accuracies, which is to be expected.

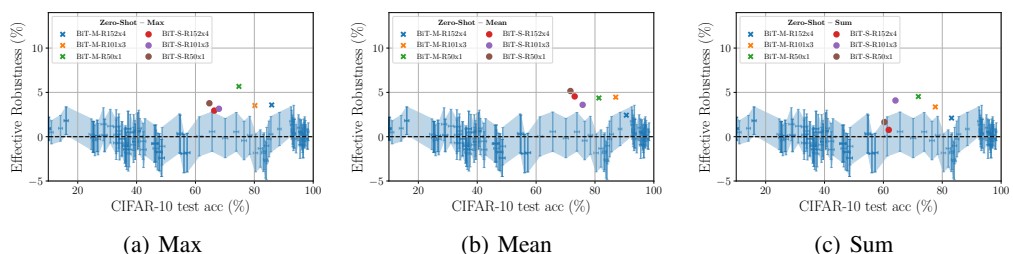

Figure 14: Zero-shot predictions using pre-trained BiT models fine-tuned to ImageNet-1k. We combine the ImageNet predictions that match the relevant CIFAR-10 class by choosing either the max, mean, or sum of the probabilities. The majority of models show some amount of ER.

All models in this section was fine-tuned using SGD with learning rate 0.001, batch size 4096, with no momentum or weight decay, and we trained for 1000 epochs.

**JFT dataset size.** Using a series of BiT-R101x3 that were pre-trained on different randomly sampled fractions of the JFT-300M dataset Sun et al. (2017), we show that the maximum ER increases with the pre-training dataset size. Since the pre-training images were randomly sampled from one dataset, it gives us some control for the dataset diversity as opposed to our experiments where we compare models trained on ImageNet-1k, ImageNet-21k and JFT. However, since the models were trained on different amounts of data, we cannot rule out any effects from the pre-training optimization on the maximum ER.

**Different pre-training datasets.** When using different pre-training datasets, we find that larger and more diverse datasets give higher maximum ER during fine-tuning on CIFAR-10.

We show the ER during fine-tuning for three different BiT model sizes in Figure 17. For R152x4 and R-101x3 we see that the BiT-L models that were pre-trained on JFT has the highest ER and the BiT-S models that were pre-trained on ImageNet-1k has the lowest. We can also make an interesting observation by looking at the two BiT-M models: The BiT-M-21k model (pre-trained on ImageNet-21k) has higher ER than the BiT-M-1k model (pre-trained on ImageNet-21k and then fine-tuned on

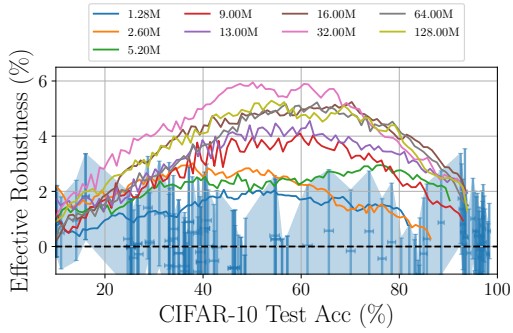

Figure 15: ER of a series of BiT-R101x3 models pre-trained on different amounts of data from JFT and fine-tuned on CIFAR-10. The maximum ER and the final accuracy increases with the JFT dataset size. The model with the maximum ER used 32 million images from JFT, and as discussed in Section 5.3, we suspect the plateau after this point is caused by the fixed number of iterations used during pre-training, which eventually serves a bottleneck for performance improvements.

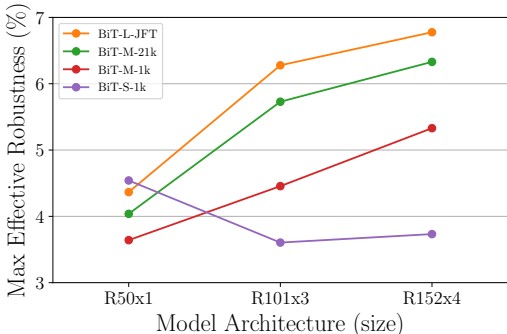

Figure 16: The maximum ER increases as a function of model architecture size for all pre-training datasets. The only exception is the BiT-S-R50x1 model that has the higest maximum ER among all the R50x1 models, and it is also higher than the BiT-S-R101x3 and BiT-S-R152x4 models.

ImageNet-1k). In other words, fine-tuning on the smaller and less diverse ImageNet-1k dataset before training on CIFAR-10 gives about a 1 percentage point drop in maximum ER.

While the observed trend that the maximum ER is higher for larger and more diverse pre-training datasets seems very robust, we observe a deviation from this trend for the BiT-S-R50x1 model. The BiT-S-R50x1 model has higher maximum ER than the BiT-L-R50x1 and BiT-M-R50x1 models. Also, it has higher ER than the comparable model with larger model architecture (BiT-S-R101x3 and BiT-S-R152x4), thus breaking yet another trend. However, we do not know if this is caused by an interesting phase transition when using the combination of the smallest pre-training dataset and the smallest model, or if it is simply caused by some part of the pre-training optimization that we do not have access to.

**Model size.** In Figure 18 we compare the ER during training for three different BiT architecture sizes for fixed pre-training dataset. For all but the BiT-S model (as discussed above), we find that the ER increases with increased model size. This is a plausible outcome since bigger models can build more expressive hidden representations.

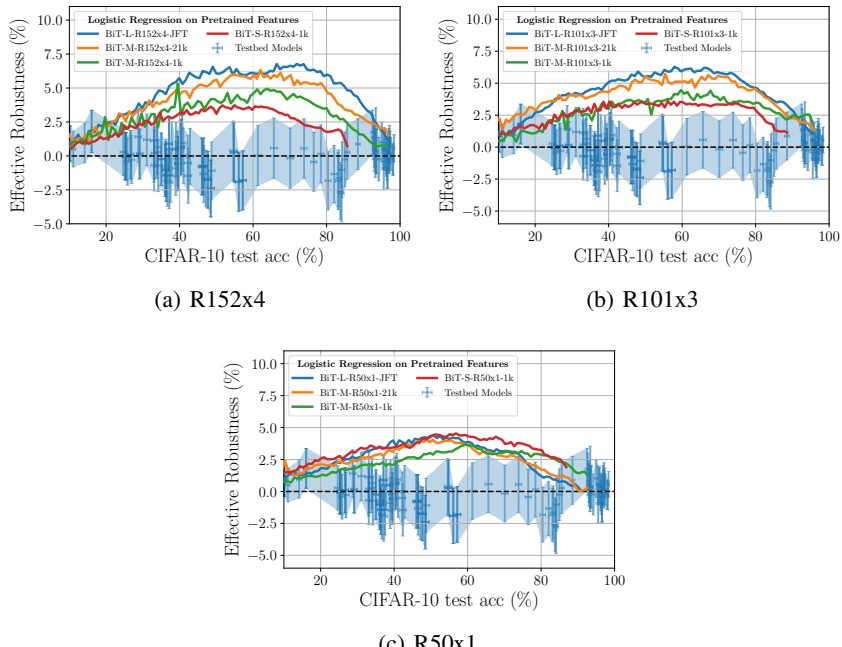

(a) R152x4        (b) R101x3

(c) R50x1

Figure 17: The ER during fine-tuning on CIFAR-10 is mostly increasing for larger pre-training dataset. We show this effect across three different architecture sizes, where each plot compares models pre-trained on different datasets.

## C.9 EXAMPLE DIFFICULTY

In this section we explore the ways in which the fine-tuning dataset affects the ER on an OOD dataset. We study a ResNet-18 model that is either randomly initialized or pre-trained on ImageNet. Both sets of models are trained with a single run for 100 epochs using batch size 64, and the pre-trained and randomly initialized models use learning rate 0.001 and 0.01, respectively.

**Fine-tuning with easy, random or hard examples.** Using the C-score Jiang et al. (2020) as a metric for example difficulty, we select 5,000 of the easiest, hardest or random examples while maintaining class balance (i.e. 500 examples from each class).

In Figure 4(c) we show the ER during training, and we make two observations: First, training on more difficult examples gives higher ER than when training on easy or random examples. Second, the models that train on only hard examples reaches a lower ID accuracy than the easy or random models.

**Phasing out easy examples during fine-tuning** Since the fine-tuning dataset affects the ER during fine-tuning, there might be a way to choose what examples from the fine-tuning dataset to train on that achieves both high ID accuracy and high ER. In Figure 19 we show the result from experiments on randomly initialized and pre-trained models where we start the fine-tuning process with the full CIFAR-10 training set, and we gradually phase out the easy examples until there are only 1000, 5000 or 10000 examples left in the last epoch of training. However, as we can see this does not seem to have any effect on the ER.

An alternative approach we attempted was switching between random, easy and hard, or first fine-tuning on all the data and switching to only the hard examples at the end. Generally, we found that when a model that reached high ID accuracy was being further fine-tuned on difficult examples, the ER does go up, but the ID accuracy also goes down. The gradual phase-out described in the previous section was the most consistent way to reach high ID accuracy while only training on hard examples by the end of training.

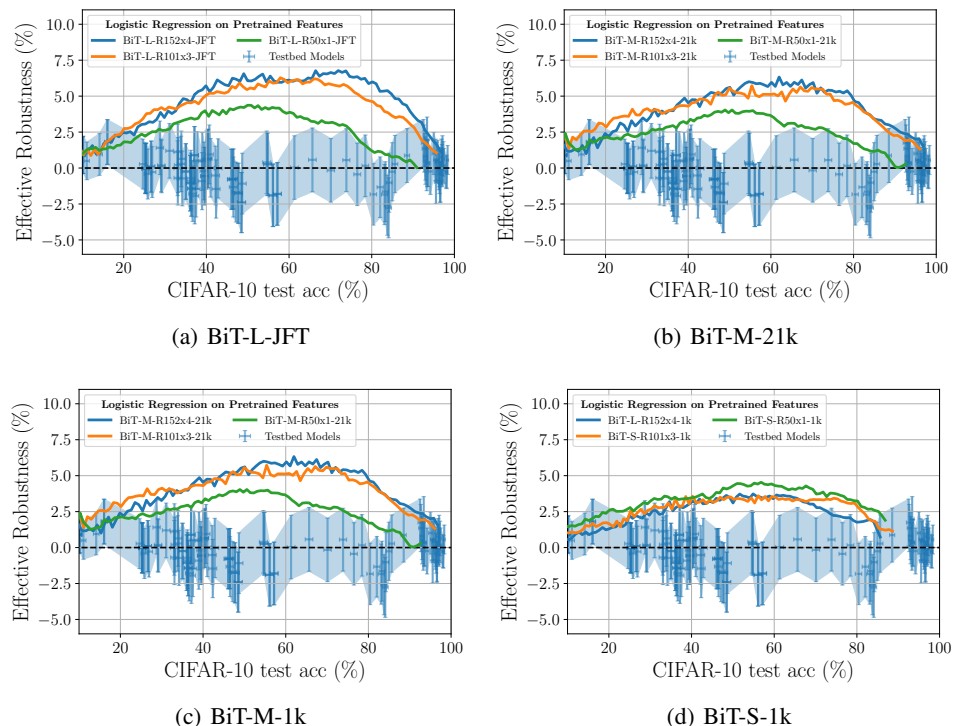

Figure 18: Larger model architectures gives higher maximum ER during fine-tuning on CIFAR-10. We show this effect for four different BiT architecture sizes for three different fixed pre-training datasets.

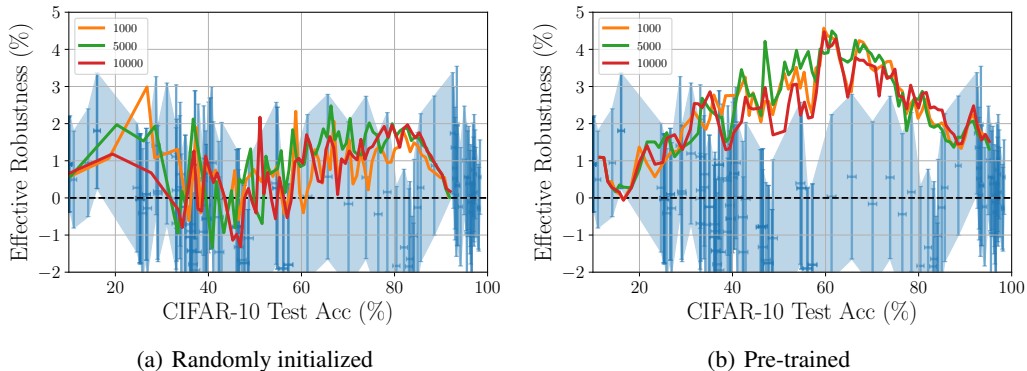

Figure 19: Since models fine-tuned on more difficult examples show more ER, we start training with the full CIFAR-10 training set. We then gradually eliminate the easiest examples (where difficulty is determined by the C-score) until we only have 1,000, 5,000 or 10,000 examples left by the last training epoch. (a) shows the ER during training for a ResNet-18 randomly initialized model, and (b) show the same model initialized with ImageNet pre-trained weights. In both cases this dataset schedule does not help maintain higher ER at high ID accuracies.

## C.10 DOMINANCE PROBABILITY

The dominance probability Mania & Sra (2020) compares the predictions of two models and is the probability that the lower accuracy model correctly classifies a given example which the higher accuracy model incorrectly classifies.

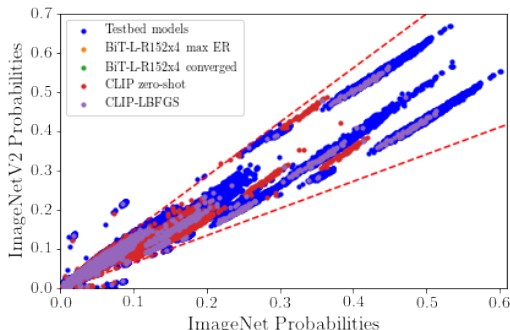

Figure 20: Triplet-probabilities between all models in our testbed in addition to two CLIP models and two BiT-L models. 99.8% of all triplet events are inside the code as defined in Mania & Sra (2020), and there is no observable violation of the distributional closeness for the effectively robust models.

**Formal definition Mania & Sra (2020).** Let $\mathcal{X}$ be a covariate space, $\mathcal{Y}$ be a discrete label space, and $\mathbb{P}$ be a probability distribution over $\mathcal{X} \times \mathcal{Y}$. Suppose model $f$ maps $\mathcal{X} \to \mathcal{Y}$ and has accuracy $\mu = \mathbb{E}_{\mathcal{P}}[\mathbb{1}(f(x) = y)]$. Given any two models $f_i$ and $f_j$ with $\mu_i \leq \mu_j$, the dominance probability is given by

$$\mathbb{P}(\{f_i(x) = y\} \cap \{f_j(x) \neq y\}) \tag{5}$$

**Pairwise dominance probability comparison.** We compute the pairwise dominance probabilities on the ImageNet validation set for the following set of models: the ImageNet testbed models, two BiT-L models (BiT-L-R152x4 early stopped at maximum ER, and the same model at end of training when the model has converged), and our zero-shot and fine-tuned CLIP models (see Appendix B.3).

Figure 21 shows a heatmap of the full set of pairwise dominance probabilities (a smaller version appears in Figure 5 in the main paper). The models are sorted by increasing ImageNet validation accuracy and the lighter colors in the heatmap correspond to models with higher dominance probabilities. Dominance probability is not symmetric, but to make the heatmap readable using either the row or column models as the more accurate model, we mirror the dominance probabilities across the $y = x$ axis.

We observe that the models with the highest dominance probabilities, the BiT-L-R152x4 Max ER model and the CLIP zero shot model, also have the highest effective robustness. However, the CLIP L-BFGS model has lower effective robustness than the BiT-L-R152x5 converged model ( 0% versus 1%, respectively), but slightly higher average dominance probability. This suggests that effective robustness alone is not the sole influencer of dominance probability, and some component of the CLIP pre-training, such as the natural language supervision, contrastive learning objective, or large and diverse pre-training dataset, must contribute to the high dominance probabilities for even the converged CLIP model.

**Distributional closeness.** To characterize distributional closeness, Mania & Sra (2020) also introduce another metric which we call the "triplet-probabilities", measuring how close the predictions between a triplet of models are (see Mania & Sra (2020) for a formal definition and discussion). In Figure 20, we show the triplet-probabilities for the same set of models as for the dominance probabilities in the last section. We observe that the effectively robust models do not violate the notion of distributional closeness as defined by the triplet probabilities, i.e. the majority of the triplet probabilities for the effectively robust models are in the same cone as the standard testbed models.

**Discussion.** Mania & Sra (2020) proved that models' accuracies are collinear under distribution shift under the assumptions of (a) low dominance probabilities and (b) distributional closeness. Conversely, if we see deviations from the linear fit, at least one of these assumtions must be violated. Oour results show that effectively robust models (which do not exhibit collinear accuracies under distribution shift) break the dominance probability assumption rather than the distributional closeness assumption. We note that ImageNet and ImageNet-V2 are explicitly constructed as to be similar, so for other OOD test sets one might expect that the distributional closeness will also be violated.

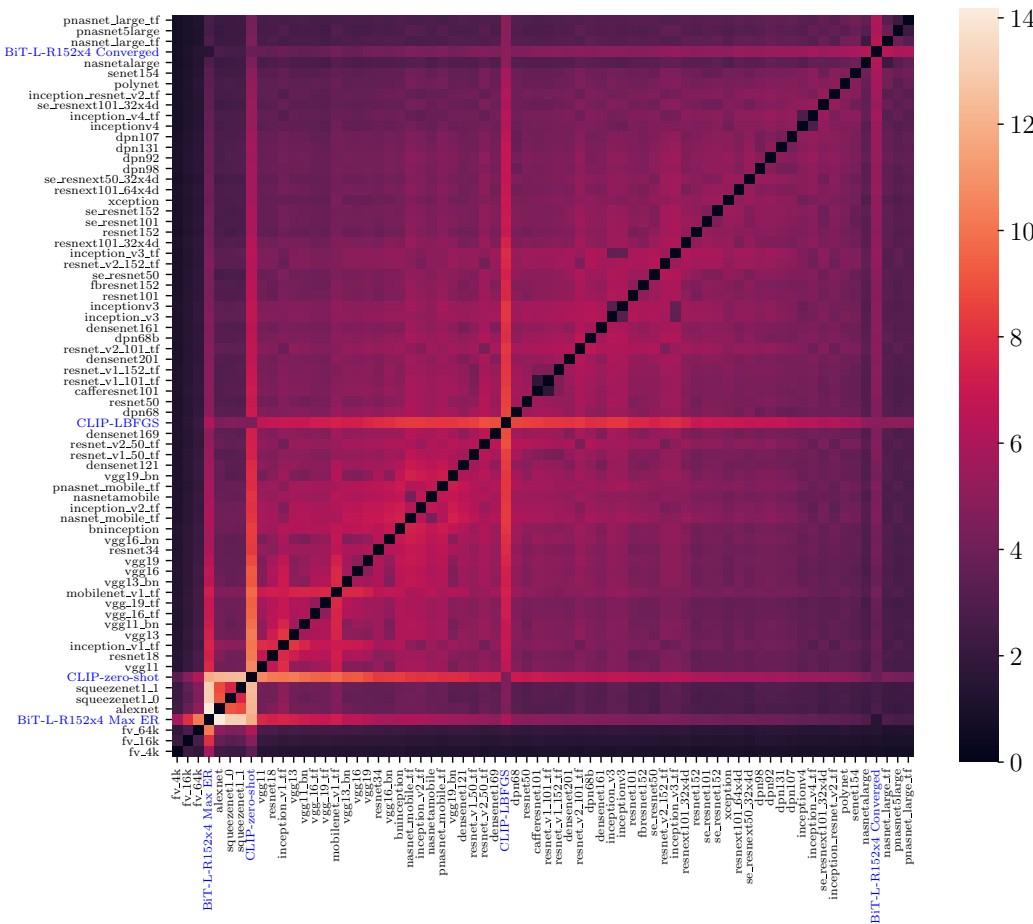

Figure 21: Pairwise dominance probabilities on ImageNet between all models in the testbed in addition to the two CLIP models, and two BiT-L models.The effectively robust models (CLIP zero-shot and BiT-L Max ER) have high dominance probability relative to all other models, which means that they make different predictions than all other models in the testbed. Even after fine-tuning of the CLIP and BiT-L model, we see from the heatmap that these models still make more different predictions than other models despite being different from the testbed models in Figure 5.

## C.11 REPLAY BUFFER: REPLAYING IMAGES FROM THE PRE-TRAINED DATASET

Our experiments using a replay buffer discussed in Section 5.5 were conducted on a ResNet-18 model that was pre-trained on ImageNet.

As a baseline, we use the pre-trained ResNet-18 model fine-tuned on only CIFAR-10 in Appendix C.3. Next, we fine-tune the pre-trained ResNet-18 model on both CIFAR-10 and ImageNet using batch size 256 and learning rate 0.001 for 100 epochs. The replay buffer model uses two prediction heads, one for ImageNet and one for CIFAR-10. For ImageNet, we kept the original 1,000 class prediction head from the pre-trained model, and the CIFAR-10 prediction head was randomly initialized. During fine-tuning, we varied the ratio of ImageNet to CIFAR-10 images to be 1, 10, or 100. The results are shown in Figure 22.

We observe that none of the replay buffer models are able to maintain high ER at high ID accuracy. This is despite the fact that such a training procedure still maintains – by design – a high accuracy on the pre-training dataset. Before any fine-tuning, the top-1 accuracy of the pre-trained ResNet-18 on the ImageNet test set is 69%. This drops to 39% during fine-tuning on CIFAR-10 when early stopped at 92% CIFAR-10 test accuracy; however, with the use of a replay buffer, the final ImageNet test accuracy is 63% at the end of fine-tuning on CIFAR-10.

This is intriguing since it demonstrates that a model that is forced to maintain high accuracy on a *different* OOD dataset during the course of fine-tuning on the target dataset does not do better on the (closer) OOD dataset of interest. For this particular setup, one might have imagined that a model that continues to perform well on ImageNet during fine-tuning on CIFAR-10 has learned more general properties of the image data manifold than it otherwise would have, and that this might be beneficial in ER on CIFAR-10.1.

In conclusion, using a replay buffer does help prevent catastrophic forgetting, but it does not prevent the ER on CIFAR-10 from disappearing towards the end of fine-tuning.

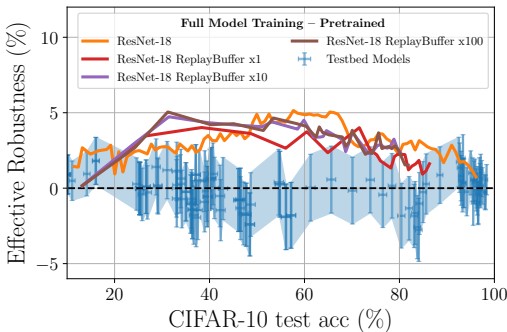

Figure 22: We study the effect of using buffer replay on the ER during fine-tuning on CIFAR-10 for a pre-trained ResNet-18 model. We compare a standard ResNet-18 to training a ResNet-18 where we see 1, 10 or 100 ImageNet batches for every CIFAR-10 batch. Despite minor differences in the x1, x10 and x100 curves, none of them are able to retain ER beyond the standard ResNet-18 model.

## C.12 CLASS MAPPING

Instead of starting from a randomly initialized prediction head when fine-tuning a pre-trained model on CIFAR-10, we can use a pre-trained prediction head if we map the ten CIFAR classes to the 1,000 ImageNet classes. For the map from CIFAR-10 to ImageNet, we use the map as defined by Darlow et al. (2018) in the construction of the CINIC-10 dataset. With this setup, we conduct several experiments to test if there is any benefit from the pre-trained prediction head in maintaining ER at high ID accuracies.

Since each CIFAR-10 class can map to multiple ImageNet classes, we use a cross-entropy loss where we select the max logit across the multi-target classes. We train the last layer of a BiT-M-R152x4_1k

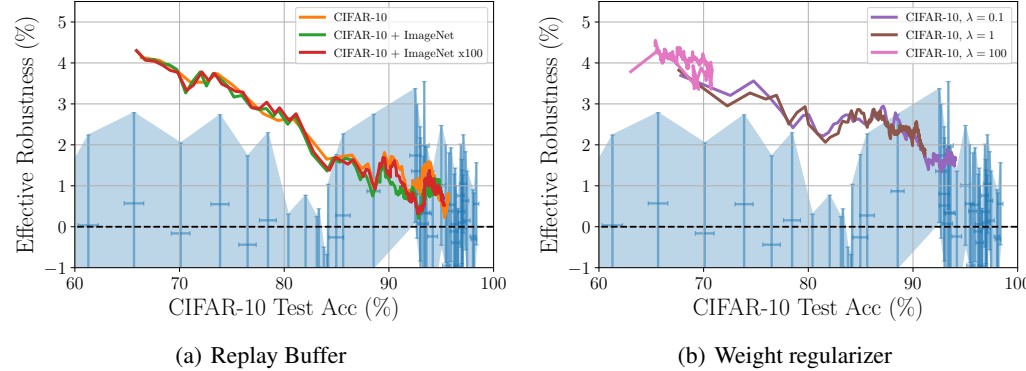

(a) Replay Buffer                    (b) Weight regularizer

Figure 23: Using a class map from CIFAR-10 to ImageNet target classes, we can fine-tune on CIFAR-10 by initializing the prediction head to the pre-trained ImageNet model. In (a) we fine-tune the last layer of a BiT-M-R152x4_1k model when training on only CIFAR-10, or using a replay buffer with 1 or 100 times as many ImageNet images. In (b) we fine-tune on only CIFAR-10, but using a L2 regularization term around the pre-trained weights, where $\lambda$ is the pre-factor of the L2 term in the loss function. However, none of them are able to maintain high ER at high ID accuracy.

model on CIFAR-10 using learning rate 0.001, batch size 32,768 for 1000 epochs. All results are shown in Figure 23.

**Replay buffer with class mapping.** Since CIFAR-10 and ImageNet now have the same target classes, we can repeat the buffer replay experiments using the same prediction head for both datasets. In Figure 23(a) we show the results from training using the same number of images from CIFAR-10 and ImageNet, and also with 100 times more ImageNet images, but we see no improvement in maintaining high ER.

**Weight regularization.** To test if any of the benefit of the pre-training comes from the pre-trained prediction head, we introduce a L2 regularization term to keep the weights close to the pre-trained weights of the prediction head. We introduce a pre-factor $\lambda$ to the L2 regularization in the loss, but we see no improvement in ER from small $\lambda$ despite a small reduction in ID accuracy relative to the results in Figure 23(a). For large $\lambda$, the L2 regularization dominates, and we do not even reach high ID accuracies.

## D    SHAPE OF THE EFFECTIVE ROBUSTNESS CURVE

In our experiments, we observe that the ER of pre-trained models starts out near zero, increases during training, reaches a peak, and then diminishes towards the end of training.

Interestingly, we find that the existence of the peak is in part caused by the linear fit in logit space and the definition of ER. To understand this, we plot the ER of the $y = x$ line (i.e. the line "Same accuracy ($y = x$)" in Figure 1 and 2) along with the ER of pre-trained BiT models in the left plot of Figure 24 (CIFAR-10) and 25 (ImageNet). The ER of the $y = x$ curve also has the same observed shape with a peak. Hence, to isolate the effect of deviating from the linear relationship from the natural shape of any ER line we also show the ER as a fraction of the accuracy gap between the linear fit and the $y = x$ line in the right plot of Figure 24 for CIFAR-10 and Figure 25 for ImageNet. We observe that these curves do not have the same maximum as observed in the ER, and instead they have a mostly decreasing trend during fine-tuning.

### D.1    MIXED CLASSIFIER

Due to the convex shape of the testbed model accuracies (without the logit scaling), one way to construct a model that has ER is to randomly sample predictions from a low accuracy testbed model with probability $\alpha$ and from a high accuracy testbed model with probability $1 - \alpha$. We call this model a mixed classifier with parameter $\alpha$. As $\alpha$ varies from 0 to 1, the accuracies on the ID and OOD

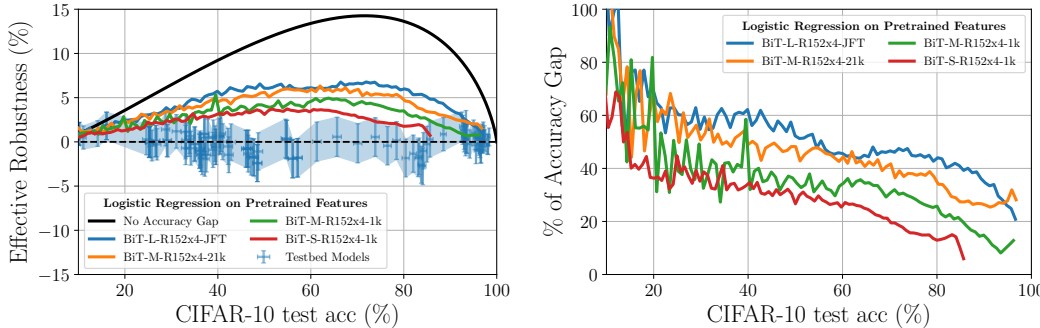

Figure 24: Left: The ER of pre-trained BiT models during fine-tuning on CIFAR-10. We include a curve indicating what the ER would be for a model that has no accuracy gap between the original and new test sets. Right: The ER as a percentage of the accuracy gap. As a percentage of the accuracy gap, the pre-trained models are slowly decreasing during fine-tuning.

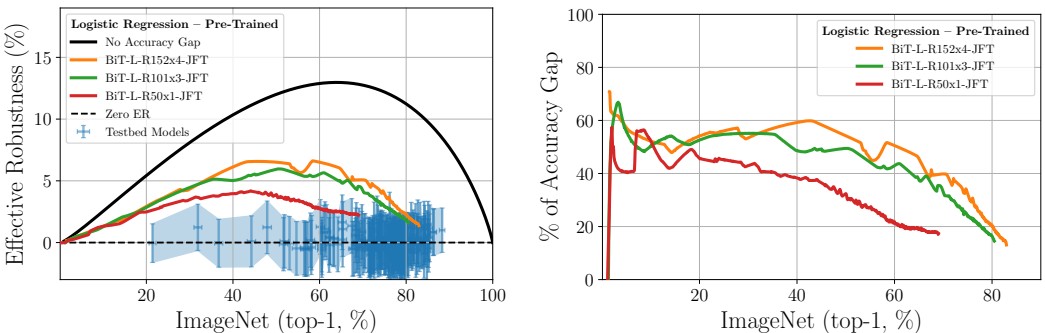

Figure 25: Same as in Figure 24 for ImageNet. Left: The ER of pre-trained BiT models during fine-tuning on ImageNet. We include a curve indicating what the ER would be for a model that has no accuracy gap between the original and new test sets. Right: The ER as a percentage of the accuracy gap. We note that as a percentage of the accuracy gap the pre-trained models are slowly decreasing during fine-tuning.

test sets traces out a straight line between the accuracy of the random and highest performing model, which will have ER due to the convexity of the logit-scaled linear fit. While mixed classifiers hold theoretical interest and should be analyzed more in future work, their use in practical applications is limited since we cannot construct a mixed classifier that has both high ER and high ID accuracy.

