# OpenReview forum: "The Evolution of Out-of-Distribution Robustness Throughout Fine-Tuning"
_ICLR.cc/2022/Conference — ICLR 2022 Submitted_

### Official Review · Reviewer_XPbn · 2021-10-23

**Correctness:** 2
**Technical Novelty And Significance:** 1
**Empirical Novelty And Significance:** 1
**Recommendation:** 3
**Confidence:** 4

**Main Review:**

I think this paper has the following strengths:

1. I think identifying models that have effective robustness and understanding their properties is an important and interesting problem. This paper has some empirical observations under this direction.

2. Enough details are included for the experiments.

3. Overall, the paper is well-written and the related work is properly discussed.

However, I think this paper has the following weaknesses:

1. My major concern is that the contribution is not very significant. The authors have some empirical observations, but those observations are not very useful and don't help us understand the problem better. For the models in the middle of fine-tuning, although they exhibit a high amount of effective robustness, the accuracy of those models on the in-distribution dataset is not high and thus such kinds of models may not be useful. Also, when the fine-tuning converges, the models have high accuracy on the in-distribution dataset but don't have effective robustness. Thus, the models obtained via fine-tuning don't have clear advantages over previous models. Besides, although the authors discuss several strategies for scaling effective robustness to the high-accuracy regime to improve the out-of-distribution accuracy, none of those methods work. So such a discussion may not be useful.

2. They only have some empirical observations, but don't have analysis for them. For example, they only show that the effective robustness generally increases throughout fine-tuning, peaks, and then gradually disappears towards the end of fine-tuning empirically, but don't analyze or explain why such a phenomenon exists. It is unclear whether such a phenomenon is general or it just exists on some datasets. It seems Figure 3 in the paper shows that such a phenomenon doesn't exist on ImageNet-R and ObjectNet when using ImageNet as in-distribution.

3. Some claims are not well supported by results. For example, the authors claim that the pre-trained models in the middle of fine-tuning, as well as zero-shot pre-trained models, represent an entire class of models that exhibit high amounts of effective robustness. I think this claim may not be true. There might be other training methods that could lead to better effective robustness and also high accuracy. The authors only explore the methods of fine-tuning and zero-shot evaluation. Thus, it is hard to claim that they represent an entire class of models that exhibit high amounts of effective robustness. The claim that the models with effective robustness make different predictions than standard models and are able to correctly classify examples that no standard models get right, is also not well supported by the results. They only select 4.8% of images that none of the testbed models get correct and show that the model that has effective robustness with the best in-distribution performance gets 10% of these examples correct. I think such results cannot support the claim that the models with effective robustness are able to correctly classify examples that no standard models get right since only 10% of these examples are predicted correctly by the model that has effective robustness with the best in-distribution performance. Also, it seems these results could not demonstrate that the models with effective robustness have prediction diversity.

4. Some observations may already be known to the community. For example, the observation that the effective robustness increases with the larger size and more diversity of the dataset seems obvious.



**Summary Of The Paper:**

In this paper, the authors conduct a thorough empirical investigation of effective robustness during fine-tuning and have several observations: 1. models pre-trained on larger datasets in the middle of fine-tuning, as well as zero-shot pre-trained models, exhibit high amounts of effective robustness, but the effective robustness vanishes at convergence; 2. the effective robustness increases with the larger size, more diversity, and higher example difficulty of the dataset; 3. models that have effective robustness make different predictions than standard models and are able to correctly classify examples that no standard models get right. Besides, they discuss several potential solutions to mitigate the problem of vanishing of effective robustness during fine-tuning, but find that none of them are able to maintain high effective robustness at high in-distribution accuracy.

**Summary Of The Review:**

I think this paper doesn't make enough contributions and the claims are not well supported by results. Also, they don't provide analysis for the observations and the observations may not be helpful in understanding the problem. Thus, I think this paper is not ready for publication.

***[Post Rebuttal]***

After reading the rebuttal, I think my major concerns still remain: the contributions are not very significant and the findings may not be useful. I still think that the models studied in this paper are not enough to represent all models that exhibit ER. The authors need to explore other kinds of models that have ER (and also have high accuracy). Thus, I keep my original rating and think the paper is not ready for publication.

---

### Official Review · Reviewer_cHRW · 2021-10-23

**Correctness:** 3
**Technical Novelty And Significance:** 2
**Empirical Novelty And Significance:** 3
**Recommendation:** 3
**Confidence:** 4

**Main Review:**

### Summary:

The paper conducts an empirical study into an interesting problem of robustness of deep models on out of distribution data. The paper finds that the pre-trained models exhibit better effective robustness during training which will disappear upon convergence of the same models.

### Pros:

The paper is well-written, easy to understand and follow along.

Most significant of all is that this paper has an extensive study of various initializations with pretrained models for vision problems.

The breadth of explorations such as pre-trained models ER during training, data set size, example difficulty,

### Cons:

are there proper bounds for the ER values, what would it really mean to have higher value, lower value etc, can you briefly explain?

ER ~ 0 for CIFAR-10 (Figure 3a) at the end of training, exactly the point at which any of the models are having the corresponding best accuracies on IN set. The only difference at that point is, the accuracy of various models is different which is already known and well-studied in literature. Similar for Imagenet, it is visible at low accuracy, and the trends are visible as the accuracy gets better similar to CIFAR. The question is, why should anyone care, if the ER is high in the middle of training at low accuracy? This is not well-justified in the current version of the paper. Also, the reasons for the peaks in ER during training are not justified, why are they intriguing? -- is it because the pre-trained models change significantly to the down-stream tasks, or something else? The random initializations don’t fluctuate that much, why not investigate these observations in detail?

In Figure 4b, why further fine-tune only the BiT-M-1k model, what happens if you further fine-tune all the models? This experiment is not a fair comparison, not all models see the same amount of data.

Again, in Figure 4c and the corresponding appendix, why would anyone use a low accuracy classifier when one knows it will perform bad on the hard to classify examples, in that case ER is not even a thing to worry in the first place, accuracy becomes the first concern. Fine, at least the ones that the classifier can classify, there is better robustness but not entirely convincing though.

This paper relies heavily on Taori et al. (2020), which seem to have a number of unresolved concerns, most important of all is that the paper is a bit short on novelty, however, the empirical study in itself is interesting.

Show the same findings hold for at least one more domain, for example, NLP.

Overall, the paper has breadth in the number of experiments and the directions that it explores without enough depth and justifications to a majority of findings.



**Summary Of The Paper:**


The paper conducts an empirical study into an interesting problem of robustness of deep models on out of distribution data. The paper finds that the pre-trained models exhibit better effective robustness during training which will disappear upon convergence of the same models.


**Summary Of The Review:**

Overall, the paper has breadth in the number of experiments and the directions that it explores without enough depth and justifications to a majority of findings. Also, the paper lacks novelty or detailed analysis of the proposed concepts. I would give it a score of 4.

---

### Official Review · Reviewer_kcNT · 2021-10-28

**Correctness:** 4
**Technical Novelty And Significance:** 2
**Empirical Novelty And Significance:** 2
**Recommendation:** 3
**Confidence:** 3

**Main Review:**

Apologies to the authors for what may sound like a rather glib judgement of this submission (and for which I acknowledge through the confidence scores below that my opinion is not absolute as I do not work directly in the space of image classification), but the results and conclusions of this paper seem remarkably obvious.  Namely, that models that have been pre-trained on a large collection of different datasets tend to lose their strengths at predicting out of distribution as they are progressively fine-tuned towards predicting a specific type of data.  And that when these models are performing in the 'effective robustness' mode the types of in sample problems they find easy (alt. hard) are different to those that models trained on the dataset at hand find easy (alt. hard).  The perils of over-fitting to a particular training set are well known and strategies to avoid this and improve generalisation are a major component of ongoing work in the machine learning (see e.g. Roger Grosse's comp sci lecture notes: https://www.cs.toronto.edu/~rgrosse/courses/csc321_2018/readings/L09%20Generalization.pdf ).  To change my mind on this point would require additional discussion by the authors to connect this work to general principles of machine learning and establish the novelty of the insights reached from these numerical experiments.

That said, my many years in research have taught me that sometimes results that seem to me to be 'remarkably obvious' are actually not so for the general audience, and that 'simple' examples demonstrating such principles can actually have a large impact and generate huge citation indices.  I mean this genuinely; not trying to be cynical here.  So for that reason I would respect the decision of other reviewers and the AEs if this paper was in fact recommended for the conference series.  Certainly, having a reference to point to for e.g. the fact that self-driving cars probably shouldn't spend too much time refining their algorithms to over-fit to a commuter's every day journey to work (this being inevitably at the expense of performance when he/she wants to take a drive in the countryside), could actually be very useful.

**Summary Of The Paper:**

In the manuscript entitled, "The Evolution of Out-of-Distribution Robustness Throughout Fine-Tuning", the authors present an empirical investigation of model exhibiting a property known as 'effective robustness'.  In particular, their focus is on how 'effective robustness' changes during fine tuning and on the characteristics of these models.

**Summary Of The Review:**

Conclusions seem 'obvious' to this reviewer, but willing to consider other opinions.

---

### Official Review · Reviewer_AB8m · 2021-11-02

**Correctness:** 4
**Technical Novelty And Significance:** 3
**Empirical Novelty And Significance:** 3
**Recommendation:** 8
**Confidence:** 3

**Main Review:**

Strengths:
The paper is very clearly written and has a thorough experimental section validating the authors' claims.
The authors have a thorough selection of experiments that validate their claims.

Weaknesses:
One weakness of this paper is that the authors do not properly define fine-tuning. While its meaning is implicit, fine-tuning is a key concept in this paper, so having a clear definition of the term seems necessary. This is especially true when considering multiple fine-tuning steps such as when fine-tuning a fine-tuned model BiT-M-21k on CIFAR-10.

The authors use a pre-trained or randomly initialized model on a large dataset, fine-tune on a smaller dataset and measure OOD accuracy on an analogous dataset to the fine-tuned dataset. It would help if the authors give some examples of when such a training procedure would be useful. Usually fine-tuning is carried out on the distribution that the model is going to be evaluated on.

An analysis of the relation between the fine-tuning dataset and the OOD test set would be useful. Right now the relationship is alluded to based on natural distribution shifts, but it's not clear how this might generalize to other types of distribution shifts.

**Summary Of The Paper:**

This paper highlights important variables impacting the effective robustness (ER) of a pre-trained, fine-tuned model.
The authors identify that increasing model size, dataset size, and example difficulty improves the ER of a pre-trained, fine-tuned model.
The experiments suggest that the zero-shot component of CLIP plays a significant role in the high value of ER CLIP achieves.
The investigation of ER on dominance probability shows that models with high ER have high dominance probability.
The authors also present a negative result showing that several reasonable approaches to maintaining high ER while fine-tuning fail.

**Summary Of The Review:**

Overall this paper is very thorough. The authors set out to investigate the role fine-tuning has on OOD robustness and they successfully identify several key variables to consider. There are many experiments in the main paper as well as in the appendix that validate their claim. This work will be very valuable to the community as it provides some insight into what variables lead to OOD robustness for pre-trained, fine-tuned models.

---

### Decision · Program_Chairs · 2022-01-20

**Decision:**

Reject

**Comment:**

Thank you for your submission to NeurIPS.  The reviewers are quite split on this paper, but some remain substantially negative even after discussion.  I'm a bit more optimistic about the paper: the observed increase then decrease in ER during fine-tuning _does_ strike me as a fundamentally interesting phenomenon, and I believe that papers that present such phenomena can be valuable contributions even without more fundamental "explanations" of the observations.  My recommendation, therefore, ultimately rests largely on the fact that I think (as is honestly evidenced by the reviews to a large degree), the presentation and contextualization of these results can be substantially improved in a future revision of the paper.  Specifically, the fact that several reviewers found the results obvious and/or not sufficiently substantiated suggests that the basic premises here are still failing to land.  I would strongly suggest revisions that clarified these points in a resubmission.